# MCMBP promotes the assembly of the MCM2–7 hetero-hexamer to ensure robust DNA replication in human cells

Yuichiro Saito[1], Venny Santosa[1], Kei-ichiro Ishiguro[2], Masato T Kanemaki[1,3]*

[1]Department of Chromosome Science, National Institute of Genetics, Research Organization of Information and Systems (ROIS), Mishima, Japan; [2]Department of Chromosome Biology, Institute of Molecular Embryology and Genetics (IMEG), Kumamoto University, Kumamoto, Japan; [3]Department of Genetics, The Graduate University for Advanced Studies (SOKENDAI), Mishima, Japan

**Abstract** The MCM2–7 hetero-hexamer is the replicative DNA helicase that plays a central role in eukaryotic DNA replication. In proliferating cells, the expression level of the MCM2–7 hexamer is kept high, which safeguards the integrity of the genome. However, how the MCM2–7 hexamer is assembled in living cells remains unknown. Here, we revealed that the MCM-binding protein (MCMBP) plays a critical role in the assembly of this hexamer in human cells. MCMBP associates with MCM3 which is essential for maintaining the level of the MCM2–7 hexamer. Acute depletion of MCMBP demonstrated that it contributes to MCM2–7 assembly using nascent MCM3. Cells depleted of MCMBP gradually ceased to proliferate because of reduced replication licensing. Under this condition, p53-positive cells exhibited arrest in the G1 phase, whereas p53-null cells entered the S phase and lost their viability because of the accumulation of DNA damage, suggesting that MCMBP is a potential target for killing p53-deficient cancers.

*For correspondence: mkanemak@nig.ac.jp

Competing interest: The authors declare that no competing interests exist.

## Editor's evaluation

This study is an important advance in the DNA replication field as the work clearly shows for the first time that the protein MCMBP is essential for the assembly of the MCM 2-7 hexamer. The levels of the hetero-hexamer are critical for genome maintenance and while more hexamers may be loaded to the DNA fiber during one cell cycle than used, such an excess is required as back-ups for replication completion requires these dormant and licensed origins.

## Introduction

The minichromosome maintenance 2–7 (MCM2–7) hexamer is a replicative helicase that plays a central role in eukaryotic DNA replication (*Bochman and Schwacha, 2009*; *Masai et al., 2010*). Its six subunits are evolutionarily related to each other and form a hetero-hexamer. In the late M to G1 phase, the MCM2–7 hexamer is loaded onto replication origins bound with ORC1–6, to form pre-replicative complexes (pre-RCs) in which the loaded MCM2–7 makes a head-to-head double-hexamer complex (*Bleichert et al., 2017*; *Evrin et al., 2009*; *Remus et al., 2009*). This reaction, which is also known as licensing, is aided by CDC6 and CDT1. Importantly, the MCM2–7 complex in pre-RCs is inactive as a helicase. Subsequently, upon the activation of two kinases, that is S-CDK and CDC7, MCM2–7 in pre-RCs is converted to the active CDC45-GINS-MCM (CMG) helicase, which becomes the core of the replisome and drives the replication forks in the S phase (*Bleichert et al., 2017*). The replication forks sometimes stall or collapse when they encounter obstacles of both intra- and extracellular origins,

leading to the accumulation of 'replication stress' (*Hills and Diffley, 2014*; *Zeman and Cimprich, 2014*). Failure or incompletion of DNA replication causes genome instability, potentially resulting in cell death, cancer development, or genetic disorder. Moreover, cancer cells experience replication stress intrinsically, which is thought to drive cancer evolution (*Hills and Diffley, 2014*). After DNA replication, unused MCM2–7 hexamers at dormant origins are removed (*Chong et al., 1995*; *Kubota et al., 1995*; *Todorov et al., 1995*). The meeting of two converging forks during replication termination leads to the ubiquitylation of MCM7 in the CMG helicase, which is recognized and extracted by the p97/CDC48 segregase (*Maric et al., 2014*; *Moreno et al., 2014*). Finally, the MCM subunits released from chromatin are recycled to form a functional MCM2–7 hexamer, for the next round of replication (*Sedlackova et al., 2020*).

Proliferating cells express high levels of the MCM2–7 hexamer, which is loaded onto origins in 3- to 10-fold excess over the level used for normal origin firing (*Edwards et al., 2002*). Only a subset of loaded MCM2–7 hexamers at origins are used for DNA replication, with the remaining complexes creating dormant origins, which are used as a backup when cells are challenged by replication stress. It has also been proposed that MCM2–7 in dormant origins works as a roadblock to slow the replication forks, thus safeguarding DNA replication (*Sedlackova et al., 2020*). In fact, metazoan cells with reduced licensed origins are sensitive to replication stress and DNA damage (*Ge et al., 2007*; *Ibarra et al., 2008*; *Woodward et al., 2006*). Moreover, mice expressing a reduced level of MCM2–7 are prone to developing cancers (*Kawabata et al., 2011*; *Kunnev et al., 2010*; *Pruitt et al., 2007*; *Shima et al., 2007*). The maintenance of a high level of MCM2–7 is important for genome maintenance in proliferating cells. The *MCM* genes are transcriptionally upregulated by E2F following growth stimulation (*Leone et al., 1998*; *Ohtani et al., 1999*). However, other than transcriptional activation for the de novo synthesis of the MCM subunits, little is known about how they are assembled into the MCM2–7 hexamer and how it is kept at high levels in proliferating cells.

The MCM-binding protein (MCMBP) was identified as a protein that associates with the MCM subunits, with the exception of MCM2, and was proposed to form an alternative MCM–MCMBP complex via the replacement of MCM2 (*Sakwe et al., 2007*). However, overexpressed MCMBP can interact with all MCM subunits with different affinities (*Nguyen et al., 2012*). MCMBP is highly conserved in eukaryotes, with the exception of its absence in budding yeast, and has an MCM-like domain with no Walker A motifs, suggesting that it shares a common ancestor with the MCM proteins. The reported phenotypes stemming from the loss of MCMBP vary among species. Sister-chromatid cohesion was defective in an ETG1/MCMBP mutant of *Arabidopsis thaliana* (*Takahashi et al., 2010*). Defects in the cell cycle and licensing were found in fission yeast after the inactivation of Mcb1/MCMBP (*Ding and Forsburg, 2011*; *Li et al., 2011*; *Santosa et al., 2013*). Moreover, MCMBP depletion in *Xenopus* egg extracts yielded a defect in the unloading of MCM2–7 from chromatin in the late S phase (*Nishiyama et al., 2011*), whereas its depletion in *Trypanosoma brucei* caused defects in gene silencing and DNA replication (*Kim, 2019*; *Kim et al., 2013*). In turn, MCMBP depletion in human cells caused nuclear deformation (*Jagannathan et al., 2012*). The recent literature reported that MCMBP protects the newly synthesized MCM subunits from degradation and promotes their nuclear transport using a nuclear localization signal (NLS) within MCMBP (*Sedlackova et al., 2020*).

Here, we report that MCMBP mainly associated with de novo synthesized MCM3 (namely, nascent MCM3) under physiological conditions and was essential for the formation of the MCM2–7 hexamer using nascent MCM3. Although the association of MCMBP with MCM3 is required for maintaining the high levels of the MCM2–7 hexamer and, thus, for supporting normal proliferation, the NLS within MCMBP was dispensable, suggesting that the main function of MCMBP lies in the promotion of the assembly of the MCM2–7 hexamer.

## Results

### MCM3 associates with MCMBP to form a subcomplex

We initially examined the MCM2–7 complexes in human cells. For this purpose, we prepared soluble and chromatin extracts from human colorectal cancer HCT116 cells (*Figure 1—figure supplement 1a*) and subsequently size fractionated the extracts using a gel filtration column (*Figure 1a*). In the chromatin extract, all MCM subunits were detected at around 600 kDa (*Figure 1a*, right), suggesting that they form a hetero-hexamer (*Prokhorova and Blow, 2000*). In the soluble extract, the MCM2–7

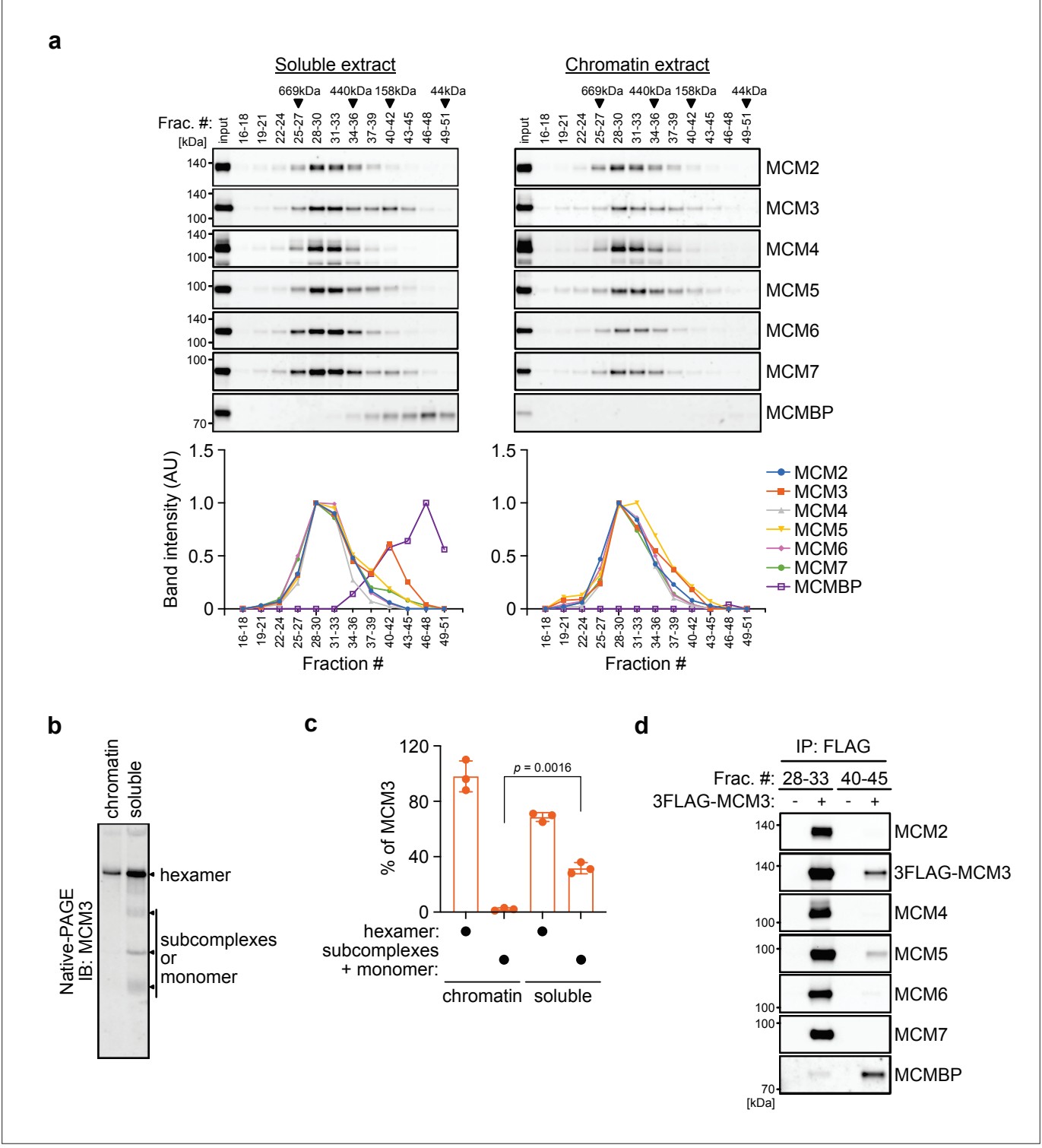

**Figure 1.** MCM3 associates with MCM-binding protein (MCMBP) in soluble extracts. (**a**) Size distribution of the MCM proteins extracted from soluble and chromatin extracts. Estimated size (using thyroglobulin, 669 kDa; ferritin, 440 kDa; aldolase, 158 kDa; and ovalbumin, 44 kDa) is indicated on the top. The intensity of each band was quantified using ImageJ and is indicated as a relative value (max = 1.0) in the graphs shown below. (**b, c**) Native PAGE for detecting MCM3 in the MCM2–7 hexamer. Proteins were extracted from chromatin and soluble fractions and subjected to native PAGE for immunoblotting with an anti-MCM3 antibody. The intensity of the MCM3 signal was measured using the Image Lab 6.0.1 software (BioRad). The means of the intensities of three independent experiments are shown as relative values (total = 100%). The error bar represents the SD. p values were determined using one-way analysis of variance (ANOVA) with Tukey's test. (**d**) Immunoprecipitation of MCM3 for detecting associating proteins in the

*Figure 1 continued on next page*

Figure 1 continued

large and small fractions. After gel filtration analysis, the indicated fractions (#28–33 or #40–45) were pooled and subjected to immunoprecipitation with an anti-FLAG antibody. Precipitated proteins were analyzed by immunoblotting using the indicated antibodies.

The online version of this article includes the following figure supplement(s) for figure 1:

**Figure supplement 1.** MCM-binding protein (MCMBP) associates with MCM3 and other MCM proteins.

subunits were detected in the similar 600 kDa fractions, but MCM3 was also detected in the fractions of smaller molecules (*Figure 1a*, left). Moreover, MCMBP was also observed in smaller fractions only in the soluble extracts, suggesting that it does not bind to chromatin and may associate with MCM3 in the soluble extract. We further examined the MCM2–7 hexamer in the soluble extract using native PAGE and detected the MCM2–7 hexamer and a subcomplex composed of MCM2/4/6/7, as reported previously (*Figure 1—figure supplement 1b*; *Prokhorova and Blow, 2000*). Interestingly, 68% of MCM3 was incorporated into the MCM2–7 hexamer, with the remainder existing as a monomer or in smaller complexes (*Figure 1b, c* and *Figure 1—figure supplement 1c*; *Tamberg et al., 2018*). To investigate whether MCM3 associates with MCMBP, a FLAG tag was fused to the endogenous MCM3 protein (*Figure 1—figure supplement 1d*). We carried out FLAG immunoprecipitation from the soluble extract after fractionation for large and small proteins, and found that MCM3 is associated with MCMBP, as well as MCM5, in the smaller fractions, as reported previously (*Figure 1d*; *Prokhorova and Blow, 2000*). These results suggest that MCM3 forms a subcomplex with MCMBP. To investigate MCM proteins interacting with MCMBP further, we used a cell line expressing 2HA-MCMBP and carried out HA immunoprecipitation after fractionation (*Figure 1—figure supplement 1e*). In the smaller fractions, MCMBP interacted with MCM3, 4, 5, 6, and 7 as well as weakly with MCM2, suggesting that MCMBP interacts with MCM monomers with different affinities as previously reported (*Kusunoki and Ishimi, 2014*; *Nguyen et al., 2012*; *Sakwe et al., 2007*). In the larger fractions, MCM4, 6, and 7 were mainly detected with MCMBP, suggesting that MCMBP also interacts with the MCM4/6/7 subcomplex (*Kimura et al., 1996*; *Prokhorova and Blow, 2000*). This immunoprecipitation data also indicated that MCMBP does not stably associate with the MCM2–7 hexamer.

## MCMBP depletion causes loss of MCM3 and MCM5 from the MCM2–7 hexamer

To investigate the effect of MCMBP depletion on the MCM2–7 hexamer, we generated a conditional mutant using the AID2 system (*Yesbolatova et al., 2020*). The MCMBP protein fused with the mini-AID (mAID) degron was rapidly degraded after the addition of an auxin analog, 5-Ph-IAA (*Figure 2a, b*). We confirmed that endogenous MCMBP fused with mAID-Clover (MCMBP-mAC) was mainly expressed in the nucleus and became undetectable after the addition of 5-Ph-IAA for 4 hr (*Figure 2—figure supplement 1a*). Even without MCMBP, we did not observe a drastic effect on the cell-cycle distribution for up to 3 days (*Figure 2—figure supplement 1b*). However, we found that the expression levels of MCM3 and MCM5 gradually decreased in the soluble extract (*Figure 2c*, soluble). Furthermore, the level of all chromatin-bound MCMs decreased, suggesting a reduced level of the MCM2–7 hexamer in the absence of MCMBP (*Figure 2c*, chromatin). We prepared a soluble extract after MCMBP depletion for 2 days and fractionated it, as shown in *Figure 1a*. We found that MCM3 was lost from the 600 kDa fractions containing the MCM2–7 hexamer and migrated only in smaller fractions (*Figure 2d*). Interestingly, MCM5 was also lost from the 600 kDa fractions. The loss of MCM3 from the MCM2–7 hexamer was confirmed by native PAGE (*Figure 2—figure supplement 1c*), and the level of the MCM2–7 hexamer became half and one-quarter on Days 1 and 2, respectively (*Figure 2—figure supplement 1d*). Considering that the doubling time of HCT116 is about 24 hr, the MCMBP-depleted cells might have grown using the MCM2–7 hexamer formed before MCMBP depletion. These results suggest that MCMBP is required for the incorporation of MCM3 and MCM5 into the MCM2–7 hexamer.

Next, we wished to determine whether the association of MCMBP with MCM3 is essential for maintaining the high levels of the MCM2–7 hexamer. For this purpose, we initially identified the interaction domain within MCMBP. The full-length MCMBP (FL) and a mutant lacking the nuclear localization signal (ΔNLS) bound to MCM3 (*Figure 3a and b*). Conversely, a mutant lacking the N terminus of MCMBP (ΔN) lost its association with MCM3. We also found that the middle and C-terminal domains

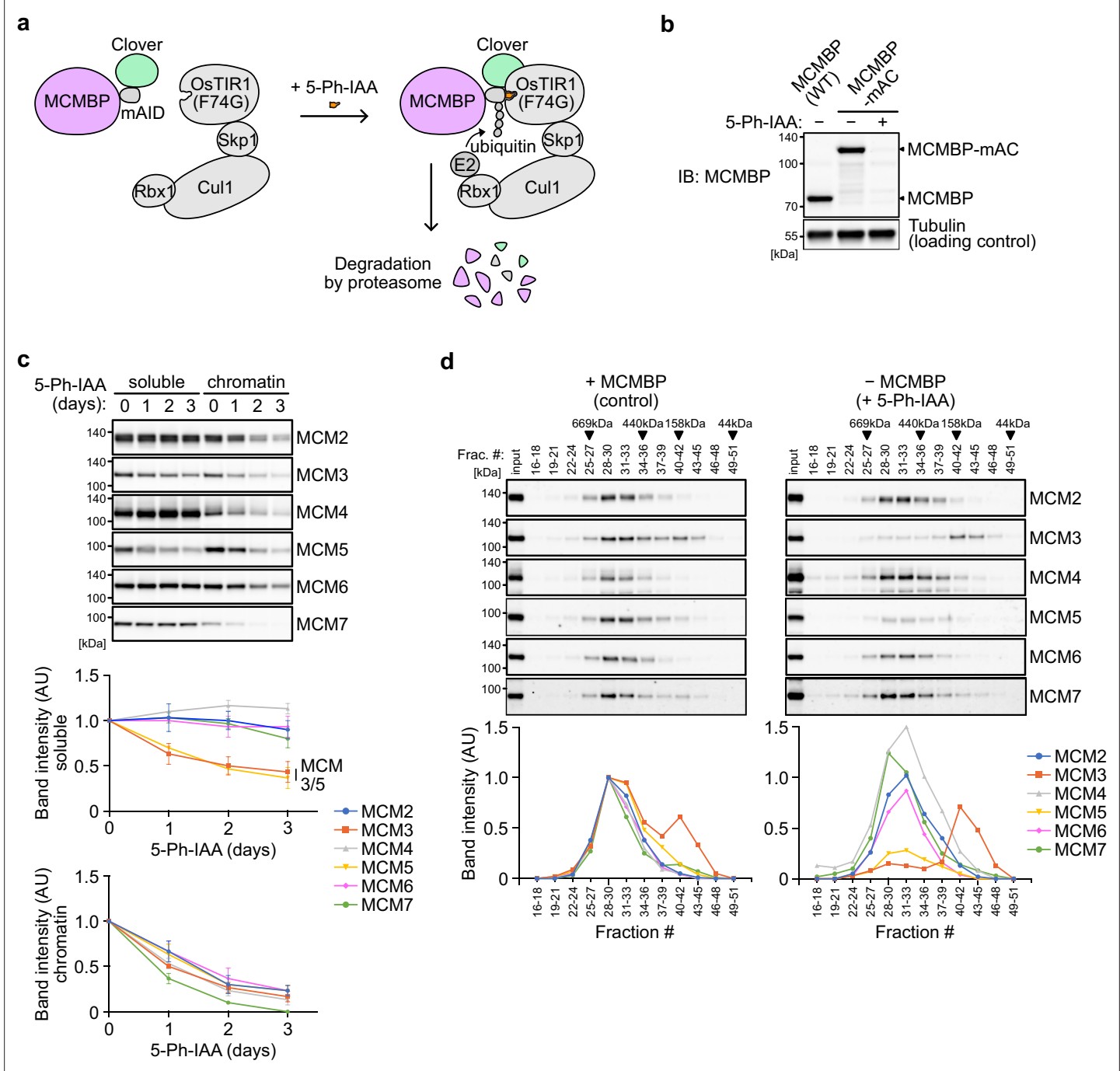

**Figure 2.** Depletion of MCM-binding protein (MCMBP) causes a defect in the formation of the MCM2–7 hexamer in the soluble extracts. (**a**) Schematic illustration of MCMBP depletion by the AID2 system. Upon 5-Ph-IAA addition, mAID-Clover (mAC)-tagged MCMBP was recognized by OsTIR1(F74G) for degradation by the proteasome. (**b**) MCMBP-mAC was induced to degrade by the addition of 1 µM 5-Ph-IAA for 4 hr. MCMBP expression was analyzed by immunoblotting. (**c**) Protein levels of soluble- and chromatin-bound MCM2–7 after MCMBP depletion. Proteins were extracted from soluble and chromatin fractions on Days 1, 2, and 3 after MCMBP depletion. The intensity of each band was measured using ImageJ and is indicated as a relative value (Day 0 = 1.0) in the graphs. Data represent the mean ± standard deviation (SD) of three independent experiments. (**d**) Size distribution of the MCM proteins in soluble extracts after MCMBP depletion. Soluble extracts were prepared 2 days after MCMBP depletion and analyzed as in *Figure 1a*. The intensity of each band was quantified using ImageJ and is indicated as a relative value (max in control = 1.0) in the graphs.

The online version of this article includes the following figure supplement(s) for figure 2:

**Figure supplement 1.** MCM-binding protein (MCMBP) depletion reduces the level of the MCM2–7 hexamer containing MCM3.

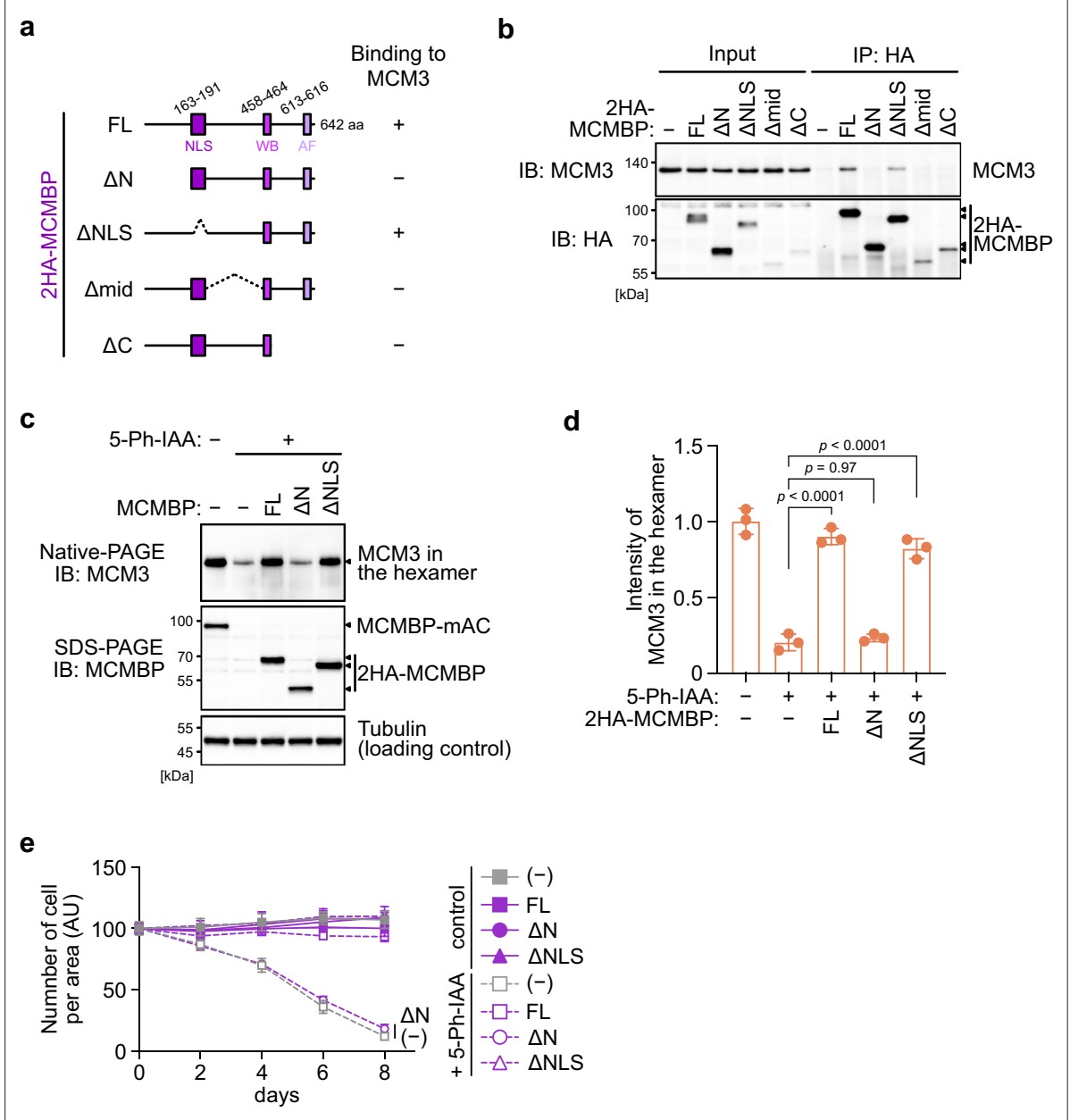

**Figure 3.** The association between MCM3 and MCM-binding protein (MCMBP) is required for maintaining the level of the MCM2–7 hexamer and supporting cell proliferation. (**a**) Illustration of the MCMBP truncations used in this study. NLS, nuclear localization signal; WB, Walker B; AF, arginine finger. (**b**) Interaction of MCMBP truncations with MCM3. HCT116 cells were transiently transfected with the HA-tagged MCMBP mutants and subjected to immunoprecipitation using an anti-HA antibody. Coprecipitated MCM3 was analyzed by immunoblotting. (**c**) Levels of the MCM2–7 hexamer in cells expressing the MCMBP mutants. The indicated MCMBP truncations were expressed in MCMBP-mAC cells; subsequently, MCMBP-mAC was depleted before preparation of the soluble extracts. Proteins were separated using native PAGE and detected using the indicated antibodies. Tubulin was used as a loading control. (**d**) The MCM2–7 hexamer in the cells expressing the indicated MCMBP truncation. The intensity of the MCM3 signal within the hexamer was quantified using the Image Lab 6.0.1 software (BioRad). The mean of the intensities of three independent experiments is indicated as a relative value (hexamer in control = 1.0). The error bar represents the SD. p values were determined using one-way analysis of variance (ANOVA) with Tukey's test. (**e**) Growth curves were obtained every 2 days after the addition of 1 µM 5-Ph-IAA. Data represent the mean ± standard deviation (SD) of three independent experiments (Day 0 = 100).

The online version of this article includes the following figure supplement(s) for figure 3:

**Figure supplement 1.** The association between MCM3 mutants and MCM-binding protein (MCMBP).

of the protein were necessary for the association, although the expression levels of these truncation mutants were lower than those of the other mutants. Subsequently, we found that the N terminus of MCM3 was bound to MCMBP (*Figure 3—figure supplement 1a, b*).

Next, we introduced an *MCMBP* transgene encoding FL, ΔN, and ΔNLS into MCMBP-mAC cells. We depleted endogenous MCMBP-mAC and assessed whether those MCMBP mutants support the retention of MCM3 in the MCM2–7 hexamer. For this purpose, we treated the cells with 5-Ph-IAA for 2 days, prepared soluble extracts and ran a native PAGE. We found that, in cells expressing MCMBP FL and ΔNLS, the level of MCM3 in the MCM2–7 hexamer was maintained (*Figure 3c, d*). Conversely, in the cells expressing MCMBP ΔN, the level of MCM3 was reduced. Subsequently, we analyzed cell proliferation after depletion of MCMBP-mAC. Cells with MCMBP-mAC depletion exhibited a gradual slowing of proliferation, which was the same even if MCMBP ΔN was expressed (*Figure 3e*). Conversely, the cells expressing MCMBP FL and ΔNLS proliferated in a manner similar to the controls after MCMBP-mAC depletion. These results indicate that the interaction of MCMBP with MCM3 is essential for maintaining MCM3 in the MCM2–7 hexamer, which is crucial for normal cell proliferation.

## MCMBP promotes the assembly of the MCM2–7 hexamer using nascent MCM3

In the absence of MCMBP, the level of the MCM2–7 hexamer decreased with time (*Figure 2—figure supplement 1c, d*), suggesting that the formation of the nascent MCM2–7 hexamer was defective. In support of this finding, *Sedlackova et al., 2020* recently reported that the stabilization and nuclear translocation of the nascent MCM subunits require MCMBP. Therefore, we reasoned that MCMBP might associate with nascent MCM3 and promote the assembly of the MCM2–7 hexamer. To monitor nascent MCM3, we introduced a transgene encoding the MCM3 protein fused with a destabilizing domain (DD) into MCMBP-mAC cells (*Figure 4—figure supplement 1a*). The DD-MCM3 protein was degraded in the absence of a DD stabilizing ligand, Shield-1, but was expressed after its addition (*Figure 4—figure supplement 1b*). We added Shield-1 to the cells with or without MCMBP and prepared soluble extracts at 0, 6, and 24 hr after its addition. Subsequently, we size fractionated the extracts and looked at nascent DD-MCM3 (*Figure 4a*). In the cells expressing MCMBP, DD-MCM3 showed two peaks around 160 and 600 kDa, respectively, at 6 hr, and the peak at 600 kDa corresponds to MCM3 within the MCM2–7 hexamer. Subsequently, the 600 kDa peak increased at 24 hr, indicating that more nascent MCM3 was incorporated, to form the MCM2–7 hexamer, following time. Conversely, in cells with MCMBP depletion, DD-MCM3 was detected around 160 kDa, and the peak at higher molecular weights was never observed. These observations are consistent with the hypothesis that MCMBP promotes the incorporation of nascent MCM3 into the MCM2–7 hexamer.

We further tested whether a functional MCM2–7 hexamer is formed with nascent MCM3. To mimic the expression timing of *MCM* genes occurring in the late G1 to S phase (*Leone et al., 1998*; *Ohtani et al., 1999*), we initially synchronized the cells at the G1 phase using lovastatin and released them in medium with or without 5-Ph-IAA (0 hr) (*Figure 4—figure supplement 1c*). Subsequently, we added Shield-1 (for expressing DD-MCM3) at 14 hr, when the cells started to enter the S phase. In the presence and absence of MCMBP, the cells progressed through the S, G2 and M phases similarly and came back to the next G1 phase at 25 hr (*Figure 4—figure supplement 1d*). We confirmed that DD-MCM3 was expressed in the S, G2 and next G1 phases (*Figure 4b, c* and *Figure 4—figure supplement 1e*), and noted that the expression level of DD-MCM3 was lower in cells with MCMBP depletion (comparison of absence/presence of MCMBP), suggesting that MCMBP protects nascent MCM3 from degradation (*Sedlackova et al., 2020*). However, nascent DD-MCM3 was still detected in the nucleus (*Figure 4b, c*). In the same condition, we extracted soluble proteins before fixation, and DD-MCM3 binding to chromatin was observed (*Figure 4d, e* and *Figure 4—figure supplement 1e*). In cells expressing MCMBP, chromatin-bound DD-MCM3 was detected only in the next G1 phase, showing that DD-MCM3 expressed after the S phase was converted to a functional MCM2–7 hexamer, which bound chromatin in the next G1 phase. Conversely, in cells with depletion of MCMBP, chromatin-bound DD-MCM3 was not detected in the next G1 phase, suggesting a defect in the formation of a functional MCM2–7 hexamer with nascent MCM3. Furthermore, we confirmed that DD-MCM3 did not bind to the other MCM subunits in the absence of MCMBP, as assessed by IP-immunoblotting (*Figure 4f*) and IP-mass spectrometry analyses (*Figure 4—figure supplement 1f*). The latter also showed that MCM3 did not interact RPA1, a subunit of RPA, without MCMBP. We interpreted these

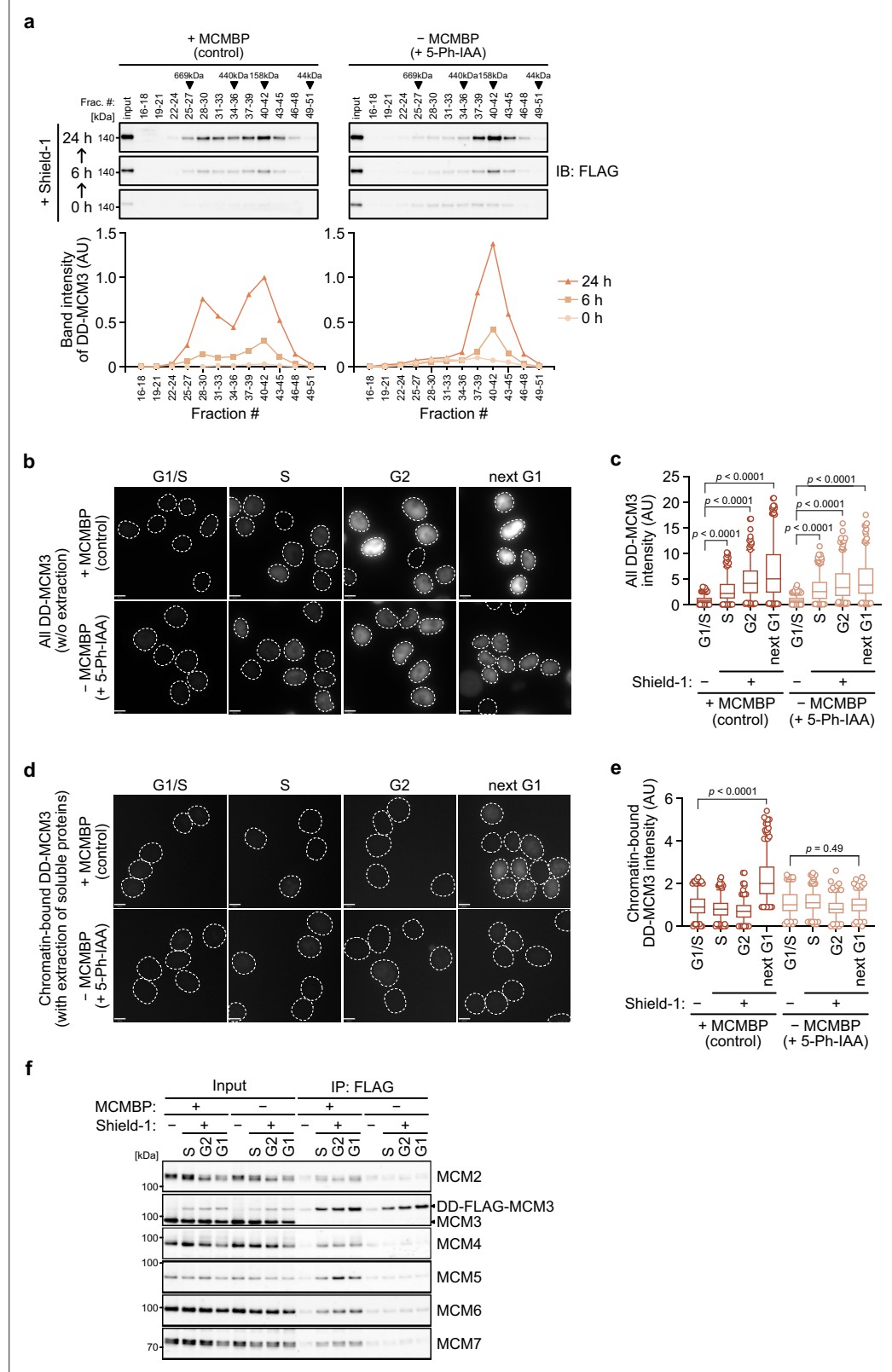

**Figure 4.** MCM-binding protein (MCMBP) promotes MCM2–7 assembly using nascent MCM3. (**a**) Size distribution of DD-FLAG-MCM3 in the presence or absence of MCMBP. Proteins were extracted and analyzed by immunoblotting at 6 or 24 hr after the addition of 0.5 µM Shield-1 with or without 1 µM 5-Ph-IAA. The intensity of each band was measured using ImageJ and is indicated as a relative value (max in control = 1.0) in the graphs. (**b**,

*Figure 4 continued on next page*

*Figure 4 continued*

**c**) Total DD-mScarletI-MCM3 after the addition of 0.5 µM Shield-1 was detected using a Delta Vision microscope. Circles with a dotted line indicate the nucleus. In the graphs, the central lines are medians. The boxes represent 25th and 75th percentiles, and the whiskers represent 5th and 95th percentiles. *n* > 200 cells per condition. AU, arbitrary units. p values were determined using one-way analysis of variance (ANOVA) with Turkey's test. (**d, e**) Chromatin-bound DD-mScarletI-MCM3 was visualized after the extraction of soluble proteins. Circles with a dotted line indicate the nucleus. The data were quantified and presented as in panel c. (**f**) Interaction of DD-FLAG-MCM3 with other MCM proteins. After immunoprecipitation using an anti-FLAG antibody, the indicated proteins were detected by immunoblotting.

The online version of this article includes the following figure supplement(s) for figure 4:

**Figure supplement 1.** MCM-binding protein (MCMBP) promotes the loading of MCM2–7 containing nascent MCM3.

**Figure supplement 2.** The nuclear transport of nascent MCM4, but not that of nascent MCM3, depends on MCM-binding protein (MCMBP).

---

data that DNA replication was defective because the functional MCM2–7 was lost in the absence of MCMBP. Based on these results, we concluded that MCMBP promotes the incorporation of nascent MCM3 into the MCM2–7 hexamer.

Among the six MCM subunits, MCM2 and MCM3 alone contain an NLS (*Kimura et al., 1996*). Recently, MCMBP, which also contains an NLS, was reported to be required for the translocation of nascent MCM4 into the nucleus (*Sedlackova et al., 2020*). To confirm this observation, we introduced DD-fused MCM4 (MCM4-DD) and induced its expression by adding Shield-1 (*Figure 4—figure supplement 2a*). Consistent with the previous report, MCM4-DD did not accumulate in the nucleus in the absence of MCMBP (*Figure 4—figure supplement 2b*), whereas DD-MCM3 accumulated in the nucleus possibly by using the NLS within MCM3 (*Figure 4—figure supplement 2c*). In the absence of MCMBP, the association of MCM3 and MCM5 with DD-MCM4 was reduced, supporting our notion that MCMBP promotes the incorporation of MCM3 into the MCM2–7 hexamer (*Figure 4—figure supplement 2d*, compare lanes 6 and 8).

## MCMBP depletion leads to loss of cell viability in P53-negative cells

The results reported above showed that MCMBP is required for the formation of a functional MCM2–7 by promoting the incorporation of nascent MCM3 (*Figure 4*). Reduced levels of the MCM2–7 hexamer lower DNA replication licensing and promote genome instabilities (*Ge et al., 2007*; *Ibarra et al., 2008*; *Kawabata et al., 2011*; *Shima et al., 2007*; *Woodward et al., 2006*). We investigated cell proliferation after MCMBP depletion and found a gradual slowing of cell growth (*Figure 5a*, MCMBP p53+/+). In these cells, chromatin-bound MCM2 was downregulated with time (*Figure 5—figure supplement 1a*), and 5-ethynyl-2′-deoxyuridine (EdU) uptake within a fixed period was increased, suggesting increased fork speed (*Figure 5—figure supplement 1b*; *Sedlackova et al., 2020*). As expected, the levels of p53 and p21 were increased on Day 5 (*Figure 5—figure supplement 1c*), suggesting that they slowed cell growth because of the cell-cycle checkpoint. In fact, on Day 8, the MCMBP-depleted cells showed a higher G1 population (*Figure 5b*, p53+/+, control). Moreover, the addition of RO-3306, which is a CDK1 inhibitor, did not deplete this G1 population (*Figure 5b*, p53+/+, + RO-3306), supporting the hypothesis that the growth of cells with MCMBP depletion was slowed by the G1 checkpoint. Therefore, we reasoned that the loss of p53 might rescue the growth defect. For this purpose, we deleted the *TP53* gene in the MCMBP-mAC background (p53−/−) and investigated cell growth after MCMBP depletion. Unexpectedly, p53−/− cells with MCMBP depletion still exhibited slowed growth, although they showed a slightly better growth than the p53+/+ cells (*Figure 5a*, MCMBP). Interestingly, p53−/− cells showed increased G2/M populations after MCMBP depletion (*Figure 5b* and p53−/−, control). This observation implies that p53−/− cells with depletion of MCMBP accumulated more DNA damage than did p53+/+ cells. In fact, p53−/− cells showed a greater number of nuclear 53BP1 foci and higher levels of H2AX phosphorylation than did p53+/+ cells (*Figure 5c* and *Figure 5—figure supplement 1d, e*).

If p53−/− cells with MCMBP depletion enter the S phase and accumulate more DNA damage, and p53+/+ cells undergo arrest in the G1 phase, we would expect that the viability of p53−/− cells should decrease more sharply than that of p53+/+ cells. To test this hypothesis, we initially depleted MCMBP

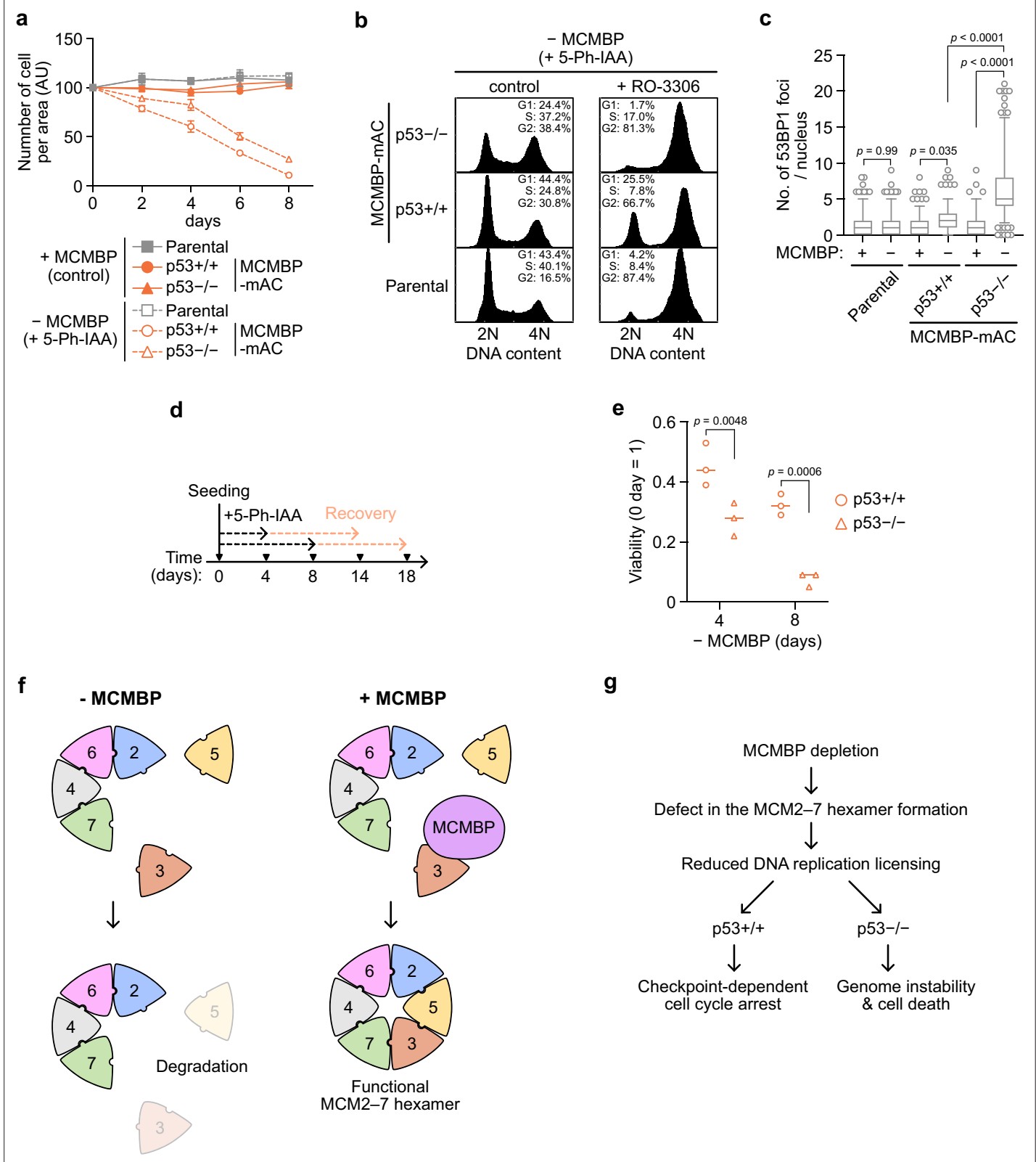

**Figure 5.** p53-negative cells lose cell viability after MCMBP depletion. (**a**) The growth of the indicated cell lines was measured every 2 days after the addition of 1 μM 5-Ph-IAA. Data represent the mean ± standard deviation (SD) of three independent experiments (Day 0 = 100). (**b**) Cell-cycle distribution on Day 8 after the addition of 1 μM 5-Ph-IAA. RO-3306 was added to the culture medium at 16 hr before fixation. (**c**) Quantification of 53BP1 foci on Day 5 after the addition of 1 μM 5-Ph-IAA. In the graphs, the central lines are medians. The boxes represent 25th and 75th percentiles, and the

*Figure 5 continued on next page*

Figure 5 continued

whiskers represent 5th and 95th percentiles. $n > 200$ cells per condition. p values were determined using one-way analysis of variance (ANOVA) with Turkey's test. (**d, e**) Cell viability after the temporal depletion of MCMBP was determined using colony formation assay. Two hundred cells were plated and cultured for 4 or 8 days in the presence of 1 μM 5-Ph-IAA, and then released into medium without 5-Ph-IAA. The plating efficiency is shown as a relative value (Day 0 = 1). Data represent the mean of three independent experiments. p values were determined using two-way ANOVA with Šídák's test. Model of MCM2–7 hexamer formation mediated by MCMBP (**f**) and the phenotypes induced after MCMBP depletion (**g**).

The online version of this article includes the following figure supplement(s) for figure 5:

**Figure supplement 1.** The p53 checkpoint prevents DNA damage accumulation after the depletion of MCM-binding protein (MCMBP).

for 4 or 8 days in the presence of 5-Ph-IAA and subsequently reexpressed MCMBP in medium without 5-Ph-IAA (*Figure 5d*). We found that p53−/− cells showed a pronounced viability loss compared with p53+/+ cells (*Figure 5e*). These results indicate that the p53-dependent checkpoint prohibited the cells from entering the S phase and protected genome integrity in the absence of MCMBP.

## Discussion

Although a recombinant MCM2–7 hexamer has successfully been reconstructed using an overexpression system based on yeast, insect, and human cells (*Jenkyn-Bedford et al., 2021*; *Rzechorzek et al., 2020*; *Yeeles et al., 2015*), little is known about how the MCM2–7 hexamer is assembled in living cells. In this study, we revealed that MCMBP played a crucial role in the assembly of the MCM2–7 hexamer in human cells. MCMBP formed a complex with MCM3 via its N-terminal region and promoted the formation of the MCM2–7 hexamer (*Figures 2 and 3*). In the absence of MCMBP, nascent MCM3 did not bind to any other of the MCM proteins, whereas the nascent MCM4 formed an MCM2/4/6/7 complex (*Figure 4* and *Figure 4—figure supplement 2*). As a result, the licensing level was gradually reduced as the cells proliferated, and the pre-existing MCM2–7 hexamer was distributed into the two daughter cells. The reduced licensing observed after MCMBP depletion caused a slow-growth phenotype in cells with and without p53 (*Figure 5*). However, the p53-negative cells became more sensitive to MCMBP depletion than did the p53-positive cells (discussed below). MCMBP is conserved in many species, including *Trypanosoma*, fission yeast, plants and mammalian cells (*Ding and Forsburg, 2011*; *Kim et al., 2013*; *Sakwe et al., 2007*; *Santosa et al., 2013*; *Takahashi et al., 2010*). We expect that the MCMBP homologs in other species might play a similar role in MCM2–7 hexamer assembly. The variable phenotypes of MCMBP mutants observed in these organisms, such as defects in sister-chromatid cohesion, cell cycle, and nuclear deformation, might stem from DNA replication with reduced licensing. Budding yeast does not possess an MCMBP ortholog, and Cdt1 strongly associates with the MCM2–7 hexamer before replication licensing and is required for the accumulation of MCM2–7 in the nucleus (*Sun et al., 2013*; *Tanaka and Diffley, 2002*). Cdt1 may also help the assembly of the hexamer in this species.

Here, we propose that MCMBP promotes the assembly of the MCM2–7 hexamer by incorporating MCM3 and MCM5 into the MCM2/4/6/7 subcomplex (*Figure 5f*). MCMBP may help the assembly of the MCM2–7 hexamer as a chaperone. Previous reports have indicated that overproduction of MCMBP reduces the levels of the MCM2–7 hexamer complex in *Xenopus* egg extracts, fission yeast, and in vitro using human proteins (*Ding and Forsburg, 2011*; *Kusunoki and Ishimi, 2014*; *Nishiyama et al., 2011*), showing that the amount of MCMBP regulates the equilibrium between the hexamer and the subcomplexes. Therefore, MCMBP may be a molecular chaperone for MCM2–7. Another possibility is that MCMBP co-operates with other chaperones for the assembly of the MCM2–7 hexamer. MCMBP was reported to associate with a prolyl-isomerase, that is FKBP5 (also known as FKBP51) (*Taipale et al., 2014*), which is involved in protein folding (*Pirkl and Buchner, 2001*). We found that MCM3 was associated with FKBP5 in an MCMBP-dependent manner (*Figure 4—figure supplement 1f*). These findings indicate that MCMBP assists the binding of FKBP5 to MCM3, resulting in MCM2–7 hexamer assembly. It has been shown that MCM3 binds to Keap1 (*Mulvaney et al., 2016*; *Tamberg et al., 2018*), which is a substrate adaptor for a CUL3 E3 ubiquitin ligase. Therefore, MCMBP may also protect nascent MCM3 from degradation via CUL3-Keap1. In fact, the expression of nascent MCM3 was reduced after MCMBP depletion (*Figure 2c*).

MCMBP was originally identified as a protein that binds to multiple MCM subunits, with the exception of MCM2 (*Sakwe et al., 2007*). However, later reports showed that MCMBP interacted with

all MCM subunits with different affinities (*Kusunoki and Ishimi, 2014*; *Nguyen et al., 2012*). We confirmed this notion in the immunoprecipitation of 2HA-MCMBP (*Figure 1—figure supplement 1e*). Notably, in the absence of MCMBP, we detected negligible changes in the levels of MCM2, 4, 6, and 7 (*Figure 2c*, soluble), indicating they formed an MCM2/4/6/7 subcomplex without MCMBP. On the contrary, the levels of MCM3 and 5 were significantly reduced in this condition (*Figure 2c*, soluble). Taking the fact that nascent MCM3 did not interact with MCM5 in the absence of MCMBP (*Figure 4—figure supplement 1f*), MCM3 and MCM5 do not form a stable subcomplex and become unstable without MCMBP (*Figure 5f*).

Among the components of the MCM2–7 hexamer, only MCM2 and MCM3 contain NLS sequences (*Kimura et al., 1996*). A previous work showed that, in fission yeast, the localization of MCM2–7 in the nucleus requires hexamer formation (*Pasion and Forsburg, 1999*). In line with this observation, nascent MCM4 did not efficiently translocate into the nucleus when MCM2–7 hexamer formation was disturbed by MCMBP depletion (*Figure 4—figure supplement 2b*). Recently, Sedlackova et al. reported that the nascent MCM subcomplexes are translocated into the nucleus with the help of the NLS signal in MCMBP, and subsequently form the MCM2–7 hexamer (*Sedlackova et al., 2020*). We found that the NLS within MCMBP was not required for MCM2–7 hexamer formation nor for maintaining cell growth (*Figure 3d, e*). This implies that, in human cells, the MCM2–7 hexamer can form in the cytoplasm, and subsequently translocate into the nucleus. However, it is also possible that MCMBP ΔNLS is associated with the localization of MCM3 in the nucleus and promotes hexamer formation in that cell compartment. In support of this hypothesis, MCMBP ΔNLS was localized both in the cytoplasm and nucleus (*Figure 3—figure supplement 1c*). Our data support the idea that the formation of the MCM2–7 hexamer can take place both in the cytoplasm and the nucleus.

When the loading of the MCM2–7 hexamer onto DNA is reduced by knockdown of ORC, Cdc6, Cdt1, or MCM2–7, the 'licensing checkpoint' arrests the cell cycle at the G1 phase (*McIntosh and Blow, 2012*). The licensing checkpoint causes the p53-dependent activation of the CDK inhibitors p21 and p27, whereas the mechanism via which the checkpoint senses the licensing level remains unknown. When MCMBP was depleted for more than 5 days, we found that the cells were arrested in the G1 phase following the activation of the p53–p21 axis (*Figure 5b* and *Figure 5—figure supplement 1c, e*). We observed that the number of 53BP1 foci (indicators of DNA double-strand breaks [DSBs]) was slightly, albeit significantly, increased after MCMBP depletion (*Figure 5c* and *Figure 5—figure supplement 1d*, compare MCMBP + and − in p53+/+, p = 0.035). Therefore, we hypothesized that, when cells enter the S phase with reduced levels of the MCM2–7 hexamer, they accumulate small amounts of DSBs. Considering that one DSB alone is sufficient to elicit cell-cycle arrest at the G1 phase (*Barlow et al., 2008*), the small number of DSBs generated after reduced licensing might be a mechanism of the licensing checkpoint.

In the absence of p53, cells with MCMBP depletion entered the S phase with reduced levels of the MCM2–7 hexamer, and then lost their viability (*Figure 5g*). A recent article showed that MCMBP accumulates at high levels in cancer cells (*Quimbaya et al., 2014*), which may be related to the over-expression of MCM2–7 observed in many cancerous tissues (*Gonzalez et al., 2005*; *Tachibana et al., 2005*). Therefore, MCMBP is a potential target for cancer chemotherapy, to kill p53-deficient cancers by allowing them to enter the S phase with an insufficient number of licensed origins (*McIntosh and Blow, 2012*), whereas normal cells are temporarily arrested in the G0 or G1 phase.

## Materials and methods
### Cell culture
HCT116 cells (ATCC, #CCL-247) were cultured in McCoy's 5A medium supplemented with 10% FBS (Sigma-Aldrich), 2 mM L-glutamine, 100 U/ml penicillin, and 100 μg/ml streptomycin at 37°C with 5% $CO_2$. Transfection was carried out as described previously (*Yesbolatova et al., 2020*). For inducing the degradation of MCMBP fused with mAID-Clover, 5-Ph-IAA (BioAcademia, 30-003) was added to the culture medium at 1 μM. For introducing the expression of DD-fused proteins, Shield-1 (Chemin-Pharma, S1) was added to the culture medium at 1 μM.

**Table 1.** Cell lines used in this study.

| Figure | Cell line |
| --- | --- |
| *Figure 1d* | HCT116 CMV-OsTIR1F74G Stag-3FLAG-MCM3 |
| *Figure 2b–d* | HCT116 CMV-OsTIR1F74G MCMBP-mAC |
| *Figure 3c–e, Figure 1—figure supplement 1e* | HCT116 CMV-OsTIR1F74G MCMBP-mAC 2HA-MCMBP FL |
| *Figure 3c–e* | HCT116 CMV-OsTIR1F74G MCMBP-mAC 2HA-MCMBPΔN |
| *Figure 3c–e* | HCT116 CMV-OsTIR1F74G MCMBP-mAC 2HA-MCMBPΔNLS |
| *Figure 4a and f* | HCT116 CMV-OsTIR1F74G MCMBP-mAC PiggyBac-EF1-DD-3FLAG-MCM3 |
| *Figure 4b–e* | HCT116 CMV-OsTIR1F74G MCMBP-mAC PiggyBac-EF1-DD-mScarletI-MCM3 |
| *Figure 5a–e* | HCT116 CMV-OsTIR1F74G MCMBP-mAC p53−/− |
| *Figure 4—figure supplement 2a, d* | HCT116 CMV-OsTIR1F74G MCMBP-mAC PiggyBac-EF1-MCM4-3FLAG-DD |
| *Figure 4—figure supplement 2b* | HCT116 CMV-OsTIR1F74G MCMBP-mAC PiggyBac-EF1-MCM4-mScarletI-DD |

## Cell lines

The original HCT116 cell line was obtained from ATCC. All HCT116 cell lines used in this study are myco-plasma negative and are listed in *Table 1*. To tag endogenous genes, we transfected a CRISPR plasmid targeting the N- or C-terminus cording region of gene of interest with a donor plasmid(s) to HCT116 cells constitutively expressing OsTIR1(F74G) as described previously (*Saito and Kanemaki, 2021*). For generating the MCMBP-mAC cells, we used a CRISPR plasmid targeting the C-terminus coding region of the *MCMBP* gene (target sequence: 5′-GTTTGCCCATTACTCTTCAT-3′) and two donors encoding mAID-Clover with a Neo/Hygro selection marker. After the selection in the presence of G418 (700 μg/ml) and hygromycin (100 μg/ml), the clones were isolated and the biallelic insertion was confirmed by genomic PCR. Subsequently, the expression of the MCMBP-mAID-Clover protein was confirmed by western blotting. For introducing MCMBP transgenes, we transfected a plasmid encoding HA-tagged MCMBP (full length or mutants) with a PiggyBac transposon plasmid to the MCMBP-mAC cells. After culturing the cells for a week in the presence of blasticidin S (10 μg/ml), the surviving cells were used. For introducing a MCM transgene fused with the DD, we transfected a plasmid encoding DD-tagged MCM3 or MCM4 with a PiggyBac transposon plasmid to the MCMBP-mAC cells. After the selection in the presence of blasticidin S (10 μg/ml), the clones were isolated and the expression was confirmed by Western blotting. For knocking out p53, we transfected a CRISPR plasmid targeting exon 3 of p53 gene (target sequence: 5′-CGGACGATATTGAACAATGG-3′) to the MCMBP-mAC cells. Clones were isolated and p53 expression loss was confirmed by western blotting.

## Plasmids

All plasmids used in this study are listed in *Table 2*.

## Preparation of cell extracts

HCT116 cells were incubated with ice-cold IP buffer (20 mM HEPES(2-[4-(2-Hydroxyethyl)-1-piperazinyl] ethanesulfonic Acid)–KOH, 150 mM KOAc, 1 mM EDTA, 5% glycerol, 5 mM Mg(OAc)$_2$, 10 mM ATP, and protease inhibitor cocktail [Merck, 11873580001]) containing 0.2% Nonidet P(NP)-40 for 2 min on ice. After low-speed centrifugation at 1200 × *g*, the supernatant was collected, further centrifuged at 16,000 × *g* and used as the soluble extract. The pellet obtained after the initial low-speed centrif-ugation was incubated in ice-cold IP buffer containing 1% NP-40 and 25–50 U of Benzonase (Merck, 70746) for 30 min on ice. After centrifugation at 16,000 × *g*, the supernatant was collected and used as the chromatin extract.

## Gel filtration

Gel filtration was carried out as described previously (*Coster et al., 2014*). Briefly, gel filtration was performed using a Superose 6 10/300 GL column (GE Healthcare) fitted to an AKTA explorer system (GE Healthcare) at a flow speed of 0.25 ml/min at 4°C. Next, 0.5 ml of soluble and chromatin extracts was applied to the column, which had been pre-equilibrated in gel filtration buffer (40 mM HEPES–KOH, 300 mM KOAc, 5 mM Mg(OAc)$_2$, 0.02% NP-40, 5% glycerol, 5 mM 2-mercaptoethanol and

**Table 2.** Plasmids used in this study.

| Figure | Plasmid |
| --- | --- |
| *Figure 1d* | MCM3-N-tagging CRISPR in pX330 |
| *Figure 1d* | Stag-3FLAG-N-MCM3 donor |
| *Figure 2b* | MCMBP-C-tagging CRISPR in pX330 |
| *Figure 2b* | MCMBP-C-tagging CRISPR in pX330 |
| *Figure 3a, b* | CMV-2HA-MCMBP FL |
| *Figure 3a, b* | CMV-2HA-MCMBPΔN |
| *Figure 3a, b* | CMV-2HA-MCMBPΔNLS |
| *Figure 3a, b* | CMV-2HA-MCMBPΔmid |
| *Figure 3a, b* | CMV-2HA-MCMBPΔC |
| *Figure 3c–e, Figure 1—figure supplement 1e* | Piggy–Bac-EF1-2HA-MCMBP FL |
| *Figure 3c–e* | PiggyBac-EF1-2HA-MCMBPΔN |
| *Figure 3c–e* | PiggyBac-EF1-2HA-MCMBPΔNLS |
| *Figure 4a, f* | PiggyBac-EF1-DD-3FLAG-MCM3 |
| *Figure 4b–e* | PiggyBac-EF1-DD-mScarletI-MCM3 |
| *Figure 5a–e* | TP53-KO CRISPR in pX330 |
| *Figure 3—figure supplement 1a, b* | TP53-KO CRISPR in pX330 |
| *Figure 3—figure supplement 1a, b* | PiggyBac-EF1-3FLAG-MCM3 FL |
| *Figure 3—figure supplement 1a, b* | PiggyBac-EF1-3FLAG-MCM3ΔN |
| *Figure 3—figure supplement 1a, b* | PiggyBac-EF1-3FLAG-MCM3ΔMCM |
| *Figure 4—figure supplement 2a, d* | PiggyBac-EF1-3FLAG-MCM3ΔC |
| *Figure 4—figure supplement 2b* | PiggyBac-EF1-MCM4-3FLAG-DD |

0.1 mM PMSF(Phenylmethylsulfonyl fluoride)). Three fractions were mixed in a tube (e.g., #1–3, #4–6, etc.) and boiled in a sodium dodecyl-sulfate (SDS) sample buffer. The sizes of proteins in the fractions were estimated using thyroglobulin (669 kDa), ferritin (440 kDa), aldolase (158 kDa), and ovalbumin (44 kDa).

## Immunoblotting
Proteins were separated using a TGX Stain-Free gel (BioRad, 4568026) or a SuperSep Ace gel (Fujifilm Wako, 197-15011), and transferred onto an Amersham Protran 0.45 µm NC membrane (Cytiva, 10600003). The membrane was incubated with the primary antibody at 4°C overnight, and subsequently with the secondary antibody at 25°C for 1 hr. Images were acquired using a ChemiDoc Touch MP imaging system (BioRad). The antibodies used in this study are listed in *Table 3*.

## Native PAGE
Proteins extracted as described above were mixed with a loading buffer (20 mM KOAc, 0.006% bromophenol blue, and 10% glycerol) and kept on ice. Proteins were separated on a SuperSep Ace gel using the Novex Tris-Glycine Native Running Buffer (Thermo, LC2672). The gel was incubated in a transfer buffer containing 0.05% SDS and subjected to protein transfer as described in Immunoblotting.

## Immunoprecipitation
Immunoprecipitation was carried out as described previously (*Hustedt et al., 2019*). To precipitate FLAG-tagged proteins, 15 µl of an anti-FLAG M2 affinity gel (Sigma-Aldrich, A2220) was added to extracts and incubated at 4°C for 2 hr with rotation. To precipitate HA-tagged proteins, 4 µg of an HA-tag antibody (MBL, M132-3) was mixed with 20 µl of Dynabeads Protein G (Invitrogen, #10,004D)

**Table 3.** Antibodies used in this study.

| Antigen | Manufacturer | Code | Method(s) | Dilution(s) |
|---|---|---|---|---|
| MCM2 | CST | 3619 | IB and IF | 1:10,000 for IB; 1:1000 for IF |
| MCM3 | Proteintech | 15597-1-AP | IB | 1:5000 |
| MCM4 | Abcam | ab4459 | IB | 1:5000 |
| MCM5 | Proteintech | 11703-1-AP | IB | 1:3000 |
| MCM6 | Proteintech | 13347-2-AP | IB | 1:5000 |
| MCM7 | Proteintech | 11225-1-AP | IB | 1:5000 |
| MCMBP | Proteintech | 19573-1-AP | IB | 1:5000 |
| FLAG | SIGMA | F1804 | IB | 1:10,000 |
| HA | MBL | M132-3 | IB | 1:5000 |
| HSP90 | Proteintech | 60,318 | IB | 1:5000 |
| Tubulin | BioRad | 12004165 | IB | 1:10,000 |
| histone H2B | Abcam | ab1790 | IB | 1:10,000 |
| p53 | MBL | K0181-3 | IB | 1:5000 |
| p21 | CST | 2946 | IB | 1:3000 |
| gH2AX | Millipore | 05-636 | IB | 1:3000 |
| anti-rabbit IgG | Abcam | ab216773 | IB | 1:3000 |
| anti-mouse IgG | Abcam | ab216772 | IB | 1:3000 |
| anti-mouse IgG | BioRad | 12004158 | IB | 1:3000 |
| anti-rabbit IgG | BioRad | 12004161 | IB | 1:3000 |
| anti-rabbit IgG | Life Technologies | A-11037 | IF | 1:500 |
| anti-mouse IgG | Life Technologies | A-21236 | IF | 1:500 |

and subsequently added to extracts. Beads were washed three times with an IP buffer containing 1% NP-40 and boiled in SDS sample buffer. The supernatant was used for immunoblotting.

## Measurement of cell proliferation

Cells were plated onto a 6-well plate and cultured in the presence or absence of 1 µM 5-Ph-IAA. Growth curves were obtained by passaging cells and measuring cell density every 2 days. Relative cell density was calculated by taking the density recorded on Day 0 as 100.

## Cell-cycle synchronization

Cell-cycle synchronization was carried out as described previously (*Natsume et al., 2016*). Briefly, cells were plated onto a 6-well plate and cultured for 2 days. Lovastatin (LKT Laboratories, M1687) was added to a final concentration of 20 µM. After 24 hr, the cells were washed once with a lovastatin-free medium and then grown in a medium containing 2 mM (±) mevalonolactone (Sigma-Aldrich, M4667) with or without 1 µM 5-Ph-IAA. The cells were collected at 14 hr (G1/S), 17 hr (S), 21 hr (G2), or 25 hr (next G1).

## Survival assay

Two hundred cells were plated onto a 6-well plate and cultured in the presence or absence of 1 µM 5-Ph-IAA for 4 or 8 days. The cells were washed three times with a 5-Ph-IAA-free medium and then grown in fresh medium for 10 days. Colonies were counted after staining with crystal violet.

## Microscopy

Cells grown in a poly-D-lysine-coated glass-bottom dish (MATTEK, P35GC-1.5-14C) were washed with phosphate-buffered saline (PBS) and fixed in 4% paraformaldehyde phosphate buffer (Fujifilm Wako,

161-20141) at 4°C for 1 hr. The cells were imaged on a Delta Vision deconvolution microscope (GE Healthcare) after DNA staining with Hoechst 33,342 (Invitrogen, H1399). Images were analyzed using the Volocity 6.3.1 software. For detecting all or chromatin-bound DD-MCM3, cells were trypsinized and incubated in PBS with or without 0.2% NP-40 for 2 min on ice. After washing with ice-cold PBS, the cells were fixed in 4% paraformaldehyde phosphate buffer at 4°C for 0.5 hr, mounted on a glass slide using CytoSpin 4 (Thermo) and stained with Hoechst 33,342. For the formation of 53BP1 foci, the cells were fixed in 4% paraformaldehyde phosphate buffer and incubated with the primary antibody at 25°C for 2 hr. After washing, the cells were incubated with the secondary antibody at 25°C for 1 hr and stained with Hoechst 33,342.

## Flow cytometry

Cells were plated onto a 6-well plate and cultured in the presence or absence of 1 µM 5-Ph-IAA. For RO-3306 treatment, RO-3306 (Sigma-Aldrich, SML0569) was added to 9 µM for 16 hr before fixation. The cells were fixed in 70% ethanol, washed with PBS and resuspended in PBS containing 1% BSA(Bovine Serum Albumin), 50 µg/ml of RNase A and 40 µg/ml of propidium iodide. For detecting MCM2, cells were incubated in PBS with 0.2% NP-40 for 2 min on ice. After washing with ice-cold PBS, the cells were fixed in 4% paraformaldehyde phosphate buffer and incubated with the primary antibody at 25°C for 2 hr. After washing, the cells were incubated with the secondary antibody at 25°C for 1 hr and resuspended in PBS containing 1% BSA, 50 µg/ml of RNase A, and 40 µg/ml of propidium iodide. For detecting DNA synthesis, cells were incubated with 10 µM EdU for 0.5 hr. EdU was visualized using the Click-iT Plus EdU Imaging Kit (Thermo, C10640), according to the manufacturer's instructions. After DNA staining at 37°C for 0.5 hr, the cells were analyzed using a BD Accuri C6 flow cytometer (BD Biosciences) and the FCS4 Express Cytometry software (DeNovo Software); 10,000 (for cell cycle) or 50,000 (for detecting MCM2 and DNA synthesis) cells were analyzed for each sample.

## Mass spectrometry

To purify FLAG-tagged DD-MCM3, $3 \times 10^7$ cells were incubated with ice-cold IP buffer containing 1% NP-40 and 150 U of Benzonase for 30 min on ice. After centrifugation at $16,000 \times g$, the supernatant was collected and subjected to the IP experiment. Next, 5 µg of a FLAG M2 antibody (Sigma-Aldrich) was mixed with 50 µl of Dynabeads Protein G, and subsequently processed through an extraction step and incubated at 4°C for 2 hr with rotation. Beads were washed three times with an IP buffer containing 1% NP-40. Bound proteins were eluted using IP buffer containing 0.25 mg/ml of 3× FLAG peptides (Sigma-Aldrich) at 25°C for 5 min. The elution step was repeated a total of three times and the eluent was collected into a tube. Proteins were precipitated with acetone by centrifugation at $16,000 \times g$, washed with methanol and dissolved in SDS sample buffer. Proteins were run for a few centimetres in SDS–polyacrylamide gel electrophoresis (PAGE). Gel bands containing proteins were excised, cut into approximately 1-mm-sized pieces and used for in-gel digestion by trypsinization before mass spectrometry. Proteins in the gel pieces were reduced with DTT (20291, Thermo Fisher Scientific), alkylated with iodoacetamide (90034, Thermo Fisher Scientific) and digested with trypsin and lysyl endopeptidase (Promega, USA) in a buffer containing 40 mM ammonium bicarbonate, pH 8.0, overnight at 37°C. The resultant peptides were analyzed on an Advance UHPLC system (AMR/Michrom Bioscience) coupled to a Q Exactive mass spectrometer (Thermo Fisher Scientific), and the raw mass spectrum was processed using Xcalibur (Thermo Fisher Scientific). The raw LC–MS/MS data were analyzed against NCBI nonredundant protein and UniProt using Proteome Discoverer version 1.4 (Thermo Fisher Scientific) using the Mascot search engine, version 2.5 (Matrix Science). A decoy database comprising either randomized or reversed sequences in the target database was used for false discovery rate (FDR) estimation, and the Percolator algorithm was used to evaluate false positives. Search results were filtered against 1% global FDR for a high confidence level.

## Acknowledgements

We thank Ms Mayumi Takahashi, Tomoko Suzuki, and Shizuoko Endo for technical support. We also thank Drs Yumiko Kurokawa, Yasuto Murayama, and Yasushi Saeki for discussion, and Dr Naoki Tani for the MS analyses. This research was supported by JSPS KAKENHI grants (JP21K15021 to YS; JP20H05396 and JP21H04719 to MTK), a JST CREST grant (JPMJCR21E6 to MTK), a research grant

from the Tokyo Biomedical Research Foundation to MTK and the Joint Usage/Research Centre for Developmental Medicine, IMEG, Kumamoto University.

## Additional information

### Funding

| Funder | Grant reference number | Author |
|---|---|---|
| Japan Society for the Promotion of Science | JP21K15021 | Yuichiro Saito |
| Japan Society for the Promotion of Science | JP20H05396 | Masato T Kanemaki |
| Japan Society for the Promotion of Science | JP21H04719 | Masato T Kanemaki |
| Japan Science and Technology Agency | JPMJCR21E6 | Masato T Kanemaki |
| Tokyo Biochemical Research Foundation | research grant | Masato T Kanemaki |
| Kumamoto University | research grant | Masato T Kanemaki |

The funders had no role in study design, data collection, and interpretation, or the decision to submit the work for publication.

### Author contributions

Yuichiro Saito, Conceptualization, Formal analysis, Funding acquisition, Investigation, Methodology, Validation, Visualization, Writing - review and editing; Venny Santosa, Investigation, Writing - review and editing; Kei-ichiro Ishiguro, Data curation, Investigation, Writing - review and editing; Masato T Kanemaki, Conceptualization, Funding acquisition, Methodology, Project administration, Resources, Supervision, Writing – original draft

### Author ORCIDs

Yuichiro Saito  http://orcid.org/0000-0001-5001-0934
Kei-ichiro Ishiguro  http://orcid.org/0000-0002-7515-1511
Masato T Kanemaki  http://orcid.org/0000-0002-7657-1649

### Decision letter and Author response

Decision letter https://doi.org/10.7554/eLife.77393.sa1
Author response https://doi.org/10.7554/eLife.77393.sa2

## Additional files

### Supplementary files

- Transparent reporting form
- Source data 1. Underlying data for all graphs.
- Source data 2. All original immunoblot images.

### Data availability

All data generated or analysed during this study are included in the manuscript and supporting files.

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
