## [Editor Report]

This study is an important advance in the DNA replication field as the work clearly shows for the first time that the protein MCMBP is essential for the assembly of the MCM 2-7 hexamer. The levels of the hetero-hexamer are critical for genome maintenance and while more hexamers may be loaded to the DNA fiber during one cell cycle than used, such an excess is required as back-ups for replication completion requires these dormant and licensed origins.

---

## [Decision Letter]

**Decision letter after peer review:**

[Editors’ note: the authors submitted for reconsideration following the decision after peer review. What follows is the decision letter after the first round of review.]

Thank you for submitting your work entitled "MCMBP maintains genome integrity by protecting the MCM subunits from degradation" for consideration by *eLife*. Your article has been reviewed by 3 peer reviewers, one of whom is a member of our Board of Reviewing Editors, and the evaluation has been overseen by a Reviewing Editor and a Senior Editor. The reviewers have opted to remain anonymous.

Our decision has been reached after consultation between the reviewers. Based on these discussions and the individual reviews below, we regret to inform you that your work will not be considered further for publication in *eLife*.

However, the reviewers recognized the potential importance of the observation that MCMBP might control the stability of the MCM2-7 helicase subunits, and indeed there was some data to support the conclusion in the paper, but it was not definitive. The reviewers noted that the conclusion that MCMBP stabilizes the MCM2-7 subunits was not demonstrated to the level that would allow acceptance of the current manuscript. If the authors can provide additional data that clearly demonstrate that MCMBP specifically stabilizes MCM2-7 subunits, then a revised manuscript will be considered, but the reviewers comments copied below will need to be addressed.

*Reviewer #1:*

The authors report that MCMBP, a protein that is known to bind to some subunits of the MCM helicase complex, is involved in maintaining the stability of some of the MCM subunits. The authors created a cell line that has auxin-inducible degradation of a tagged MCMBP protein and they find that in the presence of auxin, MCMBP disappears by 6 hr. The cells keep proliferating for 4 days with little effect, but then start to slow down compared to the absence of auxin. They also report that MCMBP is associated with subcomplexes of MCM subunits, particularly MCM5, 6 and 7, which is a potentially interesting observation. They find that upon depletion of MCMBP, some MCM subunits are unstable after a long time (from 24-96 hr). They show that the MCMBP depleted cells have a difficulty progressing through S phase.

The paper suffers from overinterpretation of the data because of the lack of any specificity of connecting MCMBP to the stability of MCM subunits. As noted below, it could be simply a loss of cell proliferation that causes selective loss of MCM subunits. The authors did not attempt to assess the levels of other pre-RC protein levels upon addition of auxin (e.g., ORC subunits, CDC6, CDT1). They did not perform critical control experiments by treating the parent HCT116-OsTIR cells with auxin and measuring any effect of auxin on these cells over the 24 to 96 hr time courses. They also do not describe the Methods well.

1. There is no information about the tagging of the MCMBP gene in the paper, except a theoretical diagram in Figure S2. There are presumably two alleles of the MCMBP gene in HCT116 cells and it was not clear if both alleles were tagged. Further, the methods describe the use of neomycin or hygromycin as selection agents, but the actual cell line used for the experiments is not described in any detail. Was it neomycin resistant, or hygromycin resistant?

2. The Methods section does not describe the HCT116 cells with OsTIR expressed and whether this expression is constitutive or induced. Only HCT116 cells are mentioned in the Methods.

3. The authors should use the appropriate figure labeling for *eLife* papers, such as Figure 1, supplement 1, etc.

4. Figure 2, panel A. What are #1 and #2? Are they independent cell lines? If so, describe them and whether they are really independent, or clines form the same isolation.

5. The results in Figure 2 show the effect of adding auxin on MCMBP protein levels, but only a 24 hr time point is shown. Since the protein has a short half life of ~ 3 hr when cycloheximide was added, the protein should disappear faster than 24 hr. A time course of protein levels after auxin treatment must be shown. It is noted that the time course is provided in Figure 3, but it should be presented first.

6. DNA staining: Figure 2 say the DNA was stained with SiR-DNA but the Methods state that Hoechst33342 stain was used for microscopy. Which is correct?

7. Figure 2B. Which antibody was used to detect MCMBP-mAC, or was the fluorescent protein detected? The Figure legend does not describe what is shown.

8. Figure 2C. It is necessary to perform the time course over a longer period than 5 days since it is not clear if MCMBP is essential, or its absence just slows proliferation. Only a longer proliferation time course will clarify this point, assuming that auxin removes all of the MCMBP-mAC, which is also not clear. 10% protein remaining would not be detected in an experiment shown in Figure 2A.

9. Figure 3. Why is MCM2, MCM4 and MCM detected in the HCT116-mAC cells (but nit MCM5) and then in RPE and HeLa cells MCM5 is added to the list. Why not detect all MCM subunits.

10. A control for the experiment in Figure 3A s needed. Treating HCT116-OsTIR cells with auxin over a time course for 48 hrs is needed since this cell line should not be sensitive to auxin. It is formally possible that auxin slows cell proliferation independent of loss of MCMBP. For example, auxin lowers MCM4 mRNA levels in HCT116-OsTIR cells (Figure S3). This is a necessary control.

11. Page 8, lines 13-15. This conclusion that MCMBP protects MCM5, 6 and 7 from degradation is far too premature based on the data in Figure 3. It could simply be that MCMBP depletion slows cell proliferation, and this indirectly affects MCM subunit levels. If this is specific, then other pre-RC protein such as ORC, CDC6 and CDT1 should not change. These must be measured.

12. A major problem is in Figure 4C. The cells were treated for 4 days with auxin (one day in the presence of lovastatin) and yet the reduction in MCM7 and MCM7 is minimal, just the same as MCM2. But in Figure 3A, a dramatic reduction is shown at 48 hours. The auxin treatment for 96 hrs in Figure 4C should show at least a reduction as great as shown at 48 he in Figure 3A, but it does not. The is clearly a difference on chromatin, but this could be a secondary effect of cell proliferation.

13. Figure 4C and 4D. Again, a control experiment using auxin treatment of HCT116-OsTIR cells for 96 hr is needed.

14. Page 10, lines 3-4 and Figure 4F. This conclusion is far too strong based on the data. The effect could be indirect since the time between MCMBP depletion upon addition of auxin (stated to be 3 hr) and the time of depletion of MCM subunits at least 24 hr for partial depletion and in Figure 4, weak depletion after 96 hr, makes it possible that the effect is due to loss of cell proliferation and not a direct effect.

*Reviewer #2:*

Significance. In addition to the individual subunits that comprise the Mcm2-7 helicase, many eukaryotes contain additional genes that encode protein that are evolutionarily related to Mcm2-7. One such proteins has been named MCMBP (Mcm binding protein) which is found I the genomes of most eukaryotes except for budding yeast. Despite its prevalence, the function of this protein has remained obscure and controversial. In this report, the authors generate and use an auxin-inducible degron to generate a conditional allele of the MCMBP gene in human cell culture to provide evidence that the role of MCMBP in this system is to prevent the inappropriate degradation of Mcm subunits in actively growing cells.

Experimental design. The unspoken assumption in this paper as well as this area of study is that since MCMBP was first identified as an MCM2-7 binding protein, MCMBP likely exerts its sole function through physical interaction with all or some of the MCM2-7 subunits. Although a reasonable conjecture, I'd be surprised that the reality is this tidy. Although to date the main effects of various types of MCMBP mutations can be rationalize as some type of replication-related defect, could this be through modulation of Mcm8 or 9? Perhaps due to its similarity to an Mcm2-7 subunit, could it be that MCMBP exerts its effect by interacting with the replication factors that Mcm2-7 normally interacts with?

The above issues extend beyond the scope of the current study. However, minimally one should ascertain if the observed defects of the MCMBP mutant are direct (i.e., require physical interaction between MCMBP and Mcm2-7) or are indirect (that binding between MCMBP and Mcm2-7 is not required). Given our current knowledge of MCMBP, one should be able generate and test MCMBP mutants that specifically do not bind Mcm subunits yet lack other collateral problems. Such data would considerably strengthen the paper and help elevate its significance from descriptive to mechanistic.

Nature of the conditional auxin-inducible MCMBP degron. In this study, the MCMBP gene was tagged with both an auxin-inducible degron construct as well as with GFP CLOVER. The expression level of the tagged MCMBP, in the presence and absence of auxin, are shown in a western blot in Figure 2A. The good news is that auxin addition results in rapid depletion of MCMBP. However, relative to the non-degron parent cell line, the expression level of the MCMBP degron construct is vastly reduced relative to the wild type protein (levels not quantified, but by eye may be <25% of the parent cell line). This problem is massaged as the authors typically show subsequent results {plus minus} auxin, not with respect to the parental cell line lacking the auxin construct.

The nature of this construct raises the potential problem that it may be producing off-target effects. In the absence of auxin, how functional is the degron mutant compared to the wild type parent cell line (e.g., growth, viability, cell cycle progression, DNA damage)? It is very difficult to evaluate the loss of MCMBP upon auxin addition unless one has carefully evaluated the original construct in the absence of auxin.

Auxin control. In addition to the above considerations, it is important to show that in the original non-degron parental cell line, that auxin addition has no effect on the stability of either MCMBP or MCM2-7. This essential control is not provided.

HeLa cell experiment. In Figures 3C and D, the key experiment using the auxin cell line (resulting in a reduction in the level of Mcm subunits after McmBP is degraded), is replicated by siRNA experiments in 2 independent cell lines (hTERT RPE and HeLa). However, only the experiment in the hTERT RPE cell line appears to replicate the basic results in the auxin cell line (i.e., that addition of the MCBP siRNA reduces the levels of Mcm subunits). To my eye, I see no reduction in the levels of MCM subunits in the parallel experiment using the HeLa cell line. This unfortunate little problem is conveniently not mentioned in the text.

*Reviewer #3:*

MCMBP is a poorly characterised protein found in most eukaryotes, which was previously shown to associate with the MCM2-7 proteins that form the core of the replicative helicase at DNA replication forks. Various functions of MCMBP have been suggested previously, such as a role in unloading MCM2-7 from chromatin in late S-phase. This manuscript provides interesting evidence to argue that MCMBP is actually a form of chaperone, which is important for proliferating cells to maintain the very high level of expression of MCM2-7. The data are generally of high quality and the study should be of considerable interest to those in the chromosome duplication and genome integrity fields.

1. The authors present evidence to indicate that MCMBP protects MCM2-7 proteins from degradation, but also "do not rule out the possibility that MCMBP helps transport nascent MCM subunits from the cytoplasm to the nucleus" (page 10, lines 19-21). Does it really take 5 days for MCM2-7 proteins to be lost from the nucleus upon rapid degradation of MCMBP-AID (as in Figure 3B), or is the effect also seen at earlier times (which might then be distinguishable from the effect of MCMBP on MCM2-7 stability)?

2. The authors suggest that MCMBP also works as a chaperone for MCM8-9 (page 11, lines 19-23), which are distantly related to MCM2-7 and function during homologous recombination. The authors mention unpublished data to show that MCMBP interacts with MCM8-9, and presumably they have already tested whether levels of MCM8-9 drop after depletion of MCMBP. If so, this would be a nice addition to the manuscript, so the authors might consider including these findings?

3. Comparing Figure 3D and 3C, the depletion by siRNA of MCMBP in HeLa cells appears to be just as efficient as in RPE1 cells, yet the effect on MCM2-7 levels is apparently much weaker in HeLa than in RPE1 cells. Do the authors have any explanation for this?

4. The authors suggest that MCMBP functions together with the chaperone FKBP51 to "promote the formation of MCM2-7 hexamers in the nucleus" (page 11, lines 4-6). Have they tested the effect of siRNA depletion of FKBP51 on the level of MCM2-7 proteins and MCMBP?

[Editors’ note: further revisions were suggested prior to acceptance, as described below.]

Thank you for resubmitting your work entitled "MCMBP promotes the assembly of the MCM2-7 hetero-hexamer to ensure robust DNA replication in human cells" for further consideration by *eLife*. Your revised article has been evaluated by Kevin Struhl (Senior Editor) and a Reviewing Editor.

The manuscript has been improved but there are some remaining issues that need to be addressed, as outlined below:

Two of the 4 reviewers thought the manuscript could be published without any revisions whilst Reviewers 1 and 4 had found similar questions that could readily be addressed with minor new additions or clarifications of the text.

1) When found in extracts the MCMBP fractionates into two forms one where Mcm3 is found and the other a high molecular weight form. It may be that the low molecular weight form has other mcm subunits aside from 3. Please address this point as described below. Further, a perhaps minor one is the nature of high molecular weight form is it MCMBP and the mcm2-7?

2) Figure 4 the supplement needs some clarification and what do you make of the fact that RPA levels decrease when MCMBP is lost?

*Reviewer #1:*

The MCM Binding Protein (MCMBP) has been implicated in many processes in cell cycle progression including DNA replication and DNA repair. The work presented in this paper shows for the first time that loss of function of MCMBP by classical cell and molecular biological methods lead to lower levels of pools of the MCM 2-7 hexamer. Moreover, MCMBP binds directly to Mcm3 and through deletions and biochemical methods in vitro, the authors show that loss of this interaction leads to lower levels of the hexamer when such defective alleles are expressed in cells. Given that the hexamer is the core of the helicase during elongation and numbers of origins are established through double hexamer formation these results will be widely impactful. Genome integrity depends upon an abundance of potential origins, particularly under stressed conditions where back -up or late origins need to finish the replication domain. The methods applied are good to establish these points but provide little direct insights into how the MCMBP actually works mechanistically in establishing these normal levels of hexamers. Is it through prevention of degradation of Mcm 3 or in an actual pathway to assembly? Nevertheless, such questions raised by the present study actually raise questions that will provide impetus for future work.

1. My only recommendation of importance for the authors is to clarify with further studies the suggestion that other Mcm's are present within the low molecular weight sub-complexes found in extracts. Figure 1 surely shows Mcm3 and also 5 and 7. There are many ways to clarify this point including a further column step or immunoprecipitation/western.

*Reviewer #2:*

MCMBP is a distant relative of the MCM2-7 proteins, which form the catalytic core of the essential DNA helicase at eukaryotic DNA replication forks. MCMBP was previously reported to associate with MCM2-7 proteins and a recent paper (Sedlackova et al, 2020, Nature, 587, 297-302) indicated that human MCMBP stabilises nascent MCM3-7 (but not MCM2) and helps translocate these helicase subunits to the nucleus. Thereby, MCMBP was proposed to be a chaperone that safeguards genome integrity by ensuring that cells contain enough MCM2-7 complexes to be loaded onto chromatin across the genome.

The manuscript by Saito et al revisits the role of MCMBP in human cells and argues that MCMBP promotes the assembly of functional MCM2-7 complexes, by association of MCMBP with MCM3, which thereby promotes the association of MCM3 with MCM2/4/6/7 (and MCM5). The authors use the recently described 'AID2' version of the auxin degron system to induce rapid degradation of MCMBP in human cells. Thereby, they show that the association of MCM3 with other MCM2-7 proteins is lost upon degradation of MCMBP, correlating with destabilisation of MCM3 and its major partner MCM5. In contrast, other MCM2-7 proteins are not destabilised upon acute depletion of MCMBP. Importantly, the NLS of MCMBP is dispensable for its role in promoting formation of the MCM2-7 hexamer. Moreover, MCMBP is shown to be required for newly synthesised MCM3 to associate with other MCM2-7 proteins, but MCMBP is not essential for the nuclear accumulation of MCM3 (that has its own NLS sequence, unlike several of the other MCM2-7 proteins that require complex formation for nuclear accumulation). Finally, the manuscript shows that depletion of MCMBP, and the associated reduction in functional MCM2-7 complexes, causes cell cycle arrest in human cells that contain p53, but DNA damage and loss of viability in cells that lack p53. These findings make an interesting contribution to our understanding of an important protein in human cells and should be of considerable interest to those in the chromosome replication and genome integrity fields.

The authors have dealt comprehensively with the points that I raised previously in relation to their original submission in 2019. The new version of the manuscript contains a large amount of new data, which are of high quality and provide strong support for the authors' conclusions. The data indicate that MCMBP associates with MCM3 and is important for MCM3 to associate with other MCM2-7 proteins, which in turn is an essential prerequisite for loading of the MCM2-7 proteins onto chromatin. The data in this revised manuscript build on the recent findings by Sedlackova et al (2020) but go considerably further and lead to important new insights. Whereas Sedlackova et al interpreted their data in terms of MCMBP promoting nuclear accumulation of nascent MCM2-7, the data in the present manuscript indicate that the key contribution of MCMBP is to promote the association of MCM3 with other MCM2-7 proteins. In the absence of MCMBP, MCM3 is still able to enter the nucleus (via its own NLS) but does not form productive MCM2-7 complexes (so nuclear accumulation of other MCM2-7 subunits is impaired).

In summary, I think that the present manuscript merits publication in eLife.

*Reviewer #3:*

Excellent manuscript revision and reviewer concerns on the original manuscript were well addressed. I support the publication of this revised manuscript in eLife.

*Reviewer #4:*

In the work described here, biochemical and cell biological approaches are used to show that MCMBP binds to MCM3 and that this complex is required for the assembly of MCM2-7 hexamer. Truncation constructs were used to define some of the important binding regions between MCMBP and MCM3. This study clarifies some of the previous data involving the role of MCMBP in supporting DNA replication and shows the importance of the protein to cell cycle control and genetic integrity. The experimental design is straightforward, and the reported observations largely appear to support the primary conclusions of the work. Whether MCMBP passively stabilizes MCM3 to protect it from degradation or actively helps catalyzes MCM2-7 hexamer formation is not resolved by the study.

Line 133 and figure 1a. The data suggest that Mcm5 and maybe Mcm7 can also be part of the Mcm3 complex with MCMBP? Figure 2d in particular indicates that Mcm5 is destabilized without MCMBP. These data should be elaborated upon in the paper and potentially incorporated into the model shown in Figure 5f.

In Figures 2d and 4a, when MCMBP was present in cells, MCM3 was found to be present in two populations, one around 600kDa and a second around 160kDa. It seems logical to presume that the two species correspond to the MCM2-7 hexamer and an MCM3-MCMBP complex (or perhaps an MCM3-MCM5-MCMBP ternary complex), respectively. However, in the absence of MCMBP (in Figures 2d and 4a), although MCM3 was present only in lower size species (fraction 37-45), the size of the lower species remained constant, similar to +MCMBP sample. One would expect that in the absence of MCMBP, MCM3 would be largely monomeric (or perhaps dimeric with MCM5) and appear in lower-size fractions. Please comment.

In Figure 1b, what are the identities of the subcomplexes? These should be resolved by mass spectrometry analysis.

Figure 4, supplement 1, panel f and accompanying text. This experiment and the data need a clearer explanation. Also, what is to be made of the loss of RPA when MCMBP is absent? And what is the identity of the gray circle next to NME1?

---

## [Author Response]

[Editors’ note: the authors resubmitted a revised version of the paper for consideration. What follows is the authors’ response to the first round of review.]

Reviewer #1:The authors report that MCMBP, a protein that is known to bind to some subunits of the MCM helicase complex, is involved in maintaining the stability of some of the MCM subunits. The authors created a cell line that has auxin-inducible degradation of a tagged MCMBP protein and they find that in the presence of auxin, MCMBP disappears by 6 hr. The cells keep proliferating for 4 days with little effect, but then start to slow down compared to the absence of auxin. They also report that MCMBP is associated with subcomplexes of MCM subunits, particularly MCM5, 6 and 7, which is a potentially interesting observation. They find that upon depletion of MCMBP, some MCM subunits are unstable after a long time (from 24-96 hr). They show that the MCMBP depleted cells have a difficulty progressing through S phase.The paper suffers from overinterpretation of the data because of the lack of any specificity of connecting MCMBP to the stability of MCM subunits. As noted below, it could be simply a loss of cell proliferation that causes selective loss of MCM subunits. The authors did not attempt to assess the levels of other pre-RC protein levels upon addition of auxin (e.g., ORC subunits, CDC6, CDT1). They did not perform critical control experiments by treating the parent HCT116-OsTIR cells with auxin and measuring any effect of auxin on these cells over the 24 to 96 hr time courses. They also do not describe the Methods well.

The new manuscript showed that MCMBP directly bound MCM3, and this interaction was required for maintaining the MCM2–7 hexamer (Figure 3). Furthermore, we revealed that MCMBP promoted the hexamer assembly of MCM2–7 using nascent MCM3 (Figure 4). Because the reduced expression of the MCM2–7 hexamer occurred before defective cell growth (compare Figure 2c and Figure 3e/5a), it was likely that the defective MCM2–7 formation caused the proliferation defect. To prove this point, we carried out rescue experiments using wild-type and mutant MCMBP (Figure 3). The wild-type MCMBP restored the hexamer formation and cell growth, whereas an MCMBP mutant lacking the interaction with MCM3 failed (Figure 3c, d, e). These data demonstrated that the proliferation defect after MCMBP depletion was caused by losing the functional MCM2–7 hexamer.

In the current manuscript, we revealed that MCMBP depletion initially affected the level

MCM3 and MCM5 in soluble extracts prepared from the cytoplasm and nucleoplasm

(Figure 2c, d). Therefore, we do not think that the replication proteins involved in preRC formation (such as ORC and CDT1) affected the loss of MCM3 and MCM5 in the cytoplasm and nucleoplasm. Please note that we are now more focusing on how the MCM2–7 assembly occurs in the current manuscript.

To avoid a side-effect caused by auxin (500 µM IAA), we employed an improved degron system, AID2, recently reported by us (Yesbolatova et al. Nat. Commun. 2020). The ligand concentration was drastically reduced using AID2. In this publication, we extensively tested the cytotoxicity of the new ligand, 5-Ph-IAA (no changes in the transcriptome and cell growth with 1 µM 5-Ph-IAA; the same experimental condition). In the current manuscript, we added this ligand in the rescue experiments (Figure 3c, d, e) and observed no problem with the MCM2–7 expression and cell growth in the MCMBP-degron cells rescued by the wild-type MCMBP transgene. Furthermore, we treated the parental HCT116 cells (without modified MCMBP) with 5-Ph-IAA for 8 days and did not find any growth defects (Figure 5a).

1. There is no information about the tagging of the MCMBP gene in the paper, except a theoretical diagram in Figure S2. There are presumably two alleles of the MCMBP gene in HCT116 cells and it was not clear if both alleles were tagged. Further, the methods describe the use of neomycin or hygromycin as selection agents, but the actual cell line used for the experiments is not described in any detail. Was it neomycin resistant, or hygromycin resistant?2. The Methods section does not describe the HCT116 cells with OsTIR expressed and whether this expression is constitutive or induced. Only HCT116 cells are mentioned in the Methods.

As we mentioned above, we changed the degron system to AID2. The new MCMBPdegron cell line continuously expressed OsTIR1(F74G) from the AAVS1 locus and MCMBP-mAID-Clover from both endogenous *MCMBP* alleles. The details were described in Methods and our protocol publication (Saito and Kanemaki, Current Protocol, 2021). All cell lines used in this study are summarized in Table S1.

3. The authors should use the appropriate figure labeling for eLife papers, such as Figure 1, supplement 1, etc.

We understand that the format can be different from the *eLife* style at the point of initial submission. Therefore, we will change the detailed format when necessary.

4. Figure 2, panel A. What are #1 and #2? Are they independent cell lines? If so, describe them and whether they are really independent, or clines form the same isolation.

We did not include any data from the previous submission.

5. The results in Figure 2 show the effect of adding auxin on MCMBP protein levels, but only a 24 hr time point is shown. Since the protein has a short half life of ~ 3 hr when cycloheximide was added, the protein should disappear faster than 24 hr. A time course of protein levels after auxin treatment must be shown. It is noted that the time course is provided in Figure 3, but it should be presented first.

We took time course samples after inducing degradation of MCMBP to monitor the expression levels of MCM2–7 (Figure 2c, Figure S2c, d). Please note that we did not add cycloheximide in these experiments. MCMBP was rapidly degraded within 4 h (Figure S2a), and the expression level of MCM2–7 gradually decreased because MCMBP was required to assemble the new complex as shown in Figure 4.

6. DNA staining: Figure 2 say the DNA was stained with SiR-DNA but the Methods state that Hoechst33342 stain was used for microscopy. Which is correct?

We used Hoechst33342 and changed the text accordingly.

7. Figure 2B. Which antibody was used to detect MCMBP-mAC, or was the fluorescent protein detected? The Figure legend does not describe what is shown.

We now include all antibody information used for detection in Table S3.

8. Figure 2C. It is necessary to perform the time course over a longer period than 5 days since it is not clear if MCMBP is essential, or its absence just slows proliferation. Only a longer proliferation time course will clarify this point, assuming that auxin removes all of the MCMBP-mAC, which is also not clear. 10% protein remaining would not be detected in an experiment shown in Figure 2A.

We extended the time course to 8 days (Figure 5a) and found that the MCMBPdepleted cells stopped growing. Depletion efficiency of MCMBP-mAC was the undetectable level (Figures 2b, S2a). We concluded MCMBP was essential for normal cell proliferation in HCT116.

9. Figure 3. Why is MCM2, MCM4 and MCM detected in the HCT116-mAC cells (but nit MCM5) and then in RPE and HeLa cells MCM5 is added to the list. Why not detect all MCM subunits.

We did not include these data using siRNA knockdown in this new submission.

10. A control for the experiment in Figure 3A s needed. Treating HCT116-OsTIR cells with auxin over a time course for 48 hrs is needed since this cell line should not be sensitive to auxin. It is formally possible that auxin slows cell proliferation independent of loss of MCMBP. For example, auxin lowers MCM4 mRNA levels in HCT116-OsTIR cells (Figure S3). This is a necessary control.12. A major problem is in Figure 4C. The cells were treated for 4 days with auxin (one day in the presence of lovastatin) and yet the reduction in MCM7 and MCM7 is minimal, just the same as MCM2. But in Figure 3A, a dramatic reduction is shown at 48 hours. The auxin treatment for 96 hrs in Figure 4C should show at least a reduction as great as shown at 48 he in Figure 3A, but it does not. The is clearly a difference on chromatin, but this could be a secondary effect of cell proliferation.13. Figure 4C and 4D. Again, a control experiment using auxin treatment of HCT116-OsTIR cells for 96 hr is needed.14. Page 10, lines 3-4 and Figure 4F. This conclusion is far too strong based on the data. The effect could be indirect since the time between MCMBP depletion upon addition of auxin (stated to be 3 hr) and the time of depletion of MCM subunits at least 24 hr for partial depletion and in Figure 4, weak depletion after 96 hr, makes it possible that the effect is due to loss of cell proliferation and not a direct effect.

Please look at our response above. The rescue experiments shown in Figure 3c, d, e should clear your concern about the side effects caused by the new ligand, 5-Ph-IAA, and the impact on the cell proliferation after MCMBP depletion.

11. Page 8, lines 13-15. This conclusion that MCMBP protects MCM5, 6 and 7 from degradation is far too premature based on the data in Figure 3. It could simply be that MCMBP depletion slows cell proliferation, and this indirectly affects MCM subunit levels. If this is specific, then other pre-RC protein such as ORC, CDC6 and CDT1 should not change. These must be measured.

Please look at our response above. Moreover, our data indicated that MCMBP protected MCM3 from degradation and promoted the MCM2–7 assembly in the cytoplasm and nucleoplasm (Figures 2c, d and S2c, d).

Reviewer #2:Significance. In addition to the individual subunits that comprise the Mcm2-7 helicase, many eukaryotes contain additional genes that encode protein that are evolutionarily related to Mcm2-7. One such proteins has been named MCMBP (Mcm binding protein) which is found I the genomes of most eukaryotes except for budding yeast. Despite its prevalence, the function of this protein has remained obscure and controversial. In this report, the authors generate and use an auxin-inducible degron to generate a conditional allele of the MCMBP gene in human cell culture to provide evidence that the role of MCMBP in this system is to prevent the inappropriate degradation of Mcm subunits in actively growing cells.Experimental design. The unspoken assumption in this paper as well as this area of study is that since MCMBP was first identified as an MCM2-7 binding protein, MCMBP likely exerts its sole function through physical interaction with all or some of the MCM2-7 subunits. Although a reasonable conjecture, I'd be surprised that the reality is this tidy. Although to date the main effects of various types of MCMBP mutations can be rationalize as some type of replication-related defect, could this be through modulation of Mcm8 or 9? Perhaps due to its similarity to an Mcm2-7 subunit, could it be that MCMBP exerts its effect by interacting with the replication factors that Mcm2-7 normally interacts with?The above issues extend beyond the scope of the current study. However, minimally one should ascertain if the observed defects of the MCMBP mutant are direct (i.e., require physical interaction between MCMBP and Mcm2-7) or are indirect (that binding between MCMBP and Mcm2-7 is not required). Given our current knowledge of MCMBP, one should be able generate and test MCMBP mutants that specifically do not bind Mcm subunits yet lack other collateral problems. Such data would considerably strengthen the paper and help elevate its significance from descriptive to mechanistic.

We removed the discussion related to MCM8/9 because we agreed that it would be beyond the scope of this manuscript. We found that the N-terminus of MCMBP interacted with MCM3 and generated an MCMBP mutant lacking this interaction (Figure 3a, b). This MCMBP∆N mutant could not rescue the hexamer formation (Figure 3c, d) and growth defect after MCMBP-mAC depletion (Figure 3e), whereas wild-type MCMBP rescued both phenotypes. These results clearly showed the MCMBP directly functioned in the MCM2–7 hexamer formation to maintain the hexamer level.

Nature of the conditional auxin-inducible MCMBP degron. In this study, the MCMBP gene was tagged with both an auxin-inducible degron construct as well as with GFP CLOVER. The expression level of the tagged MCMBP, in the presence and absence of auxin, are shown in a western blot in Figure 2A. The good news is that auxin addition results in rapid depletion of MCMBP. However, relative to the non-degron parent cell line, the expression level of the MCMBP degron construct is vastly reduced relative to the wild type protein (levels not quantified, but by eye may be <25% of the parent cell line). This problem is massaged as the authors typically show subsequent results {plus minus} auxin, not with respect to the parental cell line lacking the auxin construct.

Please note that we employed an improved degron system, AID2, in this manuscript. We did not see any leaky degradation with this new system without the inducing ligand, 5-Ph-IAA (Figure 2b). Therefore, we did not have the raised issue anymore.

The nature of this construct raises the potential problem that it may be producing off-target effects. In the absence of auxin, how functional is the degron mutant compared to the wild type parent cell line (e.g., growth, viability, cell cycle progression, DNA damage)? It is very difficult to evaluate the loss of MCMBP upon auxin addition unless one has carefully evaluated the original construct in the absence of auxin.

We compared cell proliferation, cell cycle distribution and DNA damage between the parental and MCMBP-mAC lines (p53+/+) in Figure 5a, b, c and did not observe a significant difference between the two lines. A slight increase of the G2 phase in MCMBP-mAC (p53+/+) might be because of a minor functional defect of the MCMBPmAC fusion protein.

Auxin control. In addition to the above considerations, it is important to show that in the original non-degron parental cell line, that auxin addition has no effect on the stability of either MCMBP or MCM2-7. This essential control is not provided.

As shown in Figure 5a, the parental cells were treated with 5-Ph-IAA for 8 days, and we did not observe any growth defects. As shown in Figure 3c, d, the MCMBP-mAC cells rescued by adding the wild-type *MCMBP* gene restored the MCM2–7 hexamer level. These data provided good controls, showing that the ligand did not affect the MCM2–7 level and cell growth.

HeLa cell experiment. In Figures 3C and D, the key experiment using the auxin cell line (resulting in a reduction in the level of Mcm subunits after McmBP is degraded), is replicated by siRNA experiments in 2 independent cell lines (hTERT RPE and HeLa). However, only the experiment in the hTERT RPE cell line appears to replicate the basic results in the auxin cell line (i.e., that addition of the MCBP siRNA reduces the levels of Mcm subunits). To my eye, I see no reduction in the levels of MCM subunits in the parallel experiment using the HeLa cell line. This unfortunate little problem is conveniently not mentioned in the text.

We removed these data because we could not study the assembly of MCM2–7

hexamer with slow depletion by using siRNA.

Reviewer #3:MCMBP is a poorly characterised protein found in most eukaryotes, which was previously shown to associate with the MCM2-7 proteins that form the core of the replicative helicase at DNA replication forks. Various functions of MCMBP have been suggested previously, such as a role in unloading MCM2-7 from chromatin in late S-phase. This manuscript provides interesting evidence to argue that MCMBP is actually a form of chaperone, which is important for proliferating cells to maintain the very high level of expression of MCM2-7. The data are generally of high quality and the study should be of considerable interest to those in the chromosome duplication and genome integrity fields.

We appreciate that this reviewer gave a very positive comment. We spent almost two years revising all data and now got new results showing the mechanical and biological significance of MCMBP.

1. The authors present evidence to indicate that MCMBP protects MCM2-7 proteins from degradation, but also "do not rule out the possibility that MCMBP helps transport nascent MCM subunits from the cytoplasm to the nucleus" (page 10, lines 19-21). Does it really take 5 days for MCM2-7 proteins to be lost from the nucleus upon rapid degradation of MCMBP-AID (as in Figure 3B), or is the effect also seen at earlier times (which might then be distinguishable from the effect of MCMBP on MCM2-7 stability)?

We wrote that "MCMBP might help the nuclear transport of MCM2–7" in the previous manuscript because we knew that Lukas' paper, which was under review in Nature, focused on this point (Sedlackova et al. Natrure, 2020). However, the current manuscript shows that an MCMBP mutant lacking the NLS sequence rescued the level of MCM2–7 and growth defect (Figure 3c, d, e). These data indicated that the critical function of MCMBP was to promote the assembly of MCM2–7, thus maintaining the level of MCM2–7, rather than to promote the MCM subunits to the nucleus. Furthermore, we looked at the localization of nascent MCM3 after MCMBP depletion (Figure 4b). It localized in the nucleus, possibly using NLS within MCM3, but its chromatin binding was defective (Figure 4d), supporting the idea that the hexamer formation was defective without MCMBP. In the current manuscript, we discussed what the critical function of MCMBP is and where the MCM2–7 hexamer formation occurs.

2. The authors suggest that MCMBP also works as a chaperone for MCM8-9 (page 11, lines 19-23), which are distantly related to MCM2-7 and function during homologous recombination. The authors mention unpublished data to show that MCMBP interacts with MCM8-9, and presumably they have already tested whether levels of MCM8-9 drop after depletion of MCMBP. If so, this would be a nice addition to the manuscript, so the authors might consider including these findings?

We changed all cell lines used in this study. Therefore, we have not explored the relationship between MCMBP and MCM8/9 yet. We wished to leave this issue for future research and decided not to mention MCM8/9 in the current manuscript.

3. Comparing Figure 3D and 3C, the depletion by siRNA of MCMBP in HeLa cells appears to be just as efficient as in RPE1 cells, yet the effect on MCM2-7 levels is apparently much weaker in HeLa than in RPE1 cells. Do the authors have any explanation for this?

As wrote to Reviewer 2, we removed these data because we could not study the assembly of MCM2–7 hexamer with slow depletion by using siRNA.

4. The authors suggest that MCMBP functions together with the chaperone FKBP51 to "promote the formation of MCM2-7 hexamers in the nucleus" (page 11, lines 4-6). Have they tested the effect of siRNA depletion of FKBP51 on the level of MCM2-7 proteins and MCMBP?

The current manuscript showed that the interaction between MCM3 and FKBP5 (also called FKBP51) was lost when MCMBP was depleted (Figure S4f). We wished to clarify whether FKBP5 also promoted the MCM2–7 assembly with MCMBP. For this purpose, we treated the cells with an FKBP5 inhibitor, SAFit2. However, we did not see a defect in the MCM2–7 assembly. So far, we cannot distinguish the two possibilities; FKBP5 is dispensable for the assembly, or the inhibitor is not good enough.

[Editors’ note: further revisions were suggested prior to acceptance, as described below.]

Reviewer #1 (Recommendations for the authors):The MCM Binding Protein (MCMBP) has been implicated in many processes in cell cycle progression including DNA replication and DNA repair. The work presented in this paper shows for the first time that loss of function of MCMBP by classical cell and molecular biological methods lead to lower levels of pools of the MCM 2-7 hexamer. Moreover, MCMBP binds directly to Mcm3 and through deletions and biochemical methods in vitro, the authors show that loss of this interaction leads to lower levels of the hexamer when such defective alleles are expressed in cells. Given that the hexamer is the core of the helicase during elongation and numbers of origins are established through double hexamer formation these results will be widely impactful. Genome integrity depends upon an abundance of potential origins, particularly under stressed conditions where back -up or late origins need to finish the replication domain. The methods applied are good to establish these points but provide little direct insights into how the MCMBP actually works mechanistically in establishing these normal levels of hexamers. Is it through prevention of degradation of Mcm 3 or in an actual pathway to assembly? Nevertheless, such questions raised by the present study actually raise questions that will provide impetus for future work.1. My only recommendation of importance for the authors is to clarify with further studies the suggestion that other Mcm's are present within the low molecular weight sub-complexes found in extracts. Figure 1 surely shows Mcm3 and also 5 and 7. There are many ways to clarify this point including a further column step or immunoprecipitation/western.

Please look at Figure 1—figure supplement 1e, which was added to the revised manuscript. We carried out MCMBP IP from the smaller and larger fractions (about 600 kDa and 160 kDa, respectively), and IP samples were blotted with MCM antibodies. In the smaller fractions, we detected all MCM proteins albeit MCM2 was detected less amount compared to the others. This is consistent with the previous reports, showing that MCMBP interacted with multiple MCM subunits (PMID 22540012 and 24299456). We wrote our interpretations in lines 136 to 146. Please refer to the text in the manuscript.

Reviewer #4 (Recommendations for the authors):In the work described here, biochemical and cell biological approaches are used to show that MCMBP binds to MCM3 and that this complex is required for the assembly of MCM2-7 hexamer. Truncation constructs were used to define some of the important binding regions between MCMBP and MCM3. This study clarifies some of the previous data involving the role of MCMBP in supporting DNA replication and shows the importance of the protein to cell cycle control and genetic integrity. The experimental design is straightforward, and the reported observations largely appear to support the primary conclusions of the work. Whether MCMBP passively stabilizes MCM3 to protect it from degradation or actively helps catalyzes MCM2-7 hexamer formation is not resolved by the study.Line 133 and figure 1a. The data suggest that Mcm5 and maybe Mcm7 can also be part of the Mcm3 complex with MCMBP? Figure 2d in particular indicates that Mcm5 is destabilized without MCMBP. These data should be elaborated upon in the paper and potentially incorporated into the model shown in Figure 5f.

We added a new data in Figure 1—figure supplement 1e to investigate MCMBP interactors in the smaller fractions. For this purpose, we carried out MCMBP IP from the smaller fractions and found that all MCM proteins interacted with MCMBP with different affinities, consistent with previous reports (PMID 22540012 and 24299456). Because MCM3 forms a subcomplex with either MCM5 and MCM7 (PMID 9077461 and 10644704), the smaller fractions possibly contained multiple subcomplexes, such as MCMBP/MCMs, MCM3/5 and MCM3/7. However, it appeared that MCMBP did not form a complex of MCM3/5/7/BP because MCM3 was not precipitated with MCMBP from the larger fractions (Figure 1—figure supplement 1e). We updated the model in Figure 5f for clarification.

In Figures 2d and 4a, when MCMBP was present in cells, MCM3 was found to be present in two populations, one around 600kDa and a second around 160kDa. It seems logical to presume that the two species correspond to the MCM2-7 hexamer and an MCM3-MCMBP complex (or perhaps an MCM3-MCM5-MCMBP ternary complex), respectively. However, in the absence of MCMBP (in Figures 2d and 4a), although MCM3 was present only in lower size species (fraction 37-45), the size of the lower species remained constant, similar to +MCMBP sample. One would expect that in the absence of MCMBP, MCM3 would be largely monomeric (or perhaps dimeric with MCM5) and appear in lower-size fractions. Please comment.

We understand your point. However, MCM3 monomer (91 kDa) and MCMBP/MCM3 subcomplex (73 kDa + 91 kDa) cannot be distinguished using Superose 6 column because we cannot achieve fine resolutions at the lower molecular weight. This is possibly why MCM3 in the smaller fractions were detected similarly with and without MCMBP in Figure 4a.

In Figure 1b, what are the identities of the subcomplexes? These should be resolved by mass spectrometry analysis.

Please look at Figure 1—figure supplement 1b. Among the three bands, the smallest band might be MCM3 monomer. The other two appears to colocalize with MCMBP. However, it is difficult to say what these subcomplex are. We are sorry that we cannot inform you conclusively.

Figure 4, supplement 1, panel f and accompanying text. This experiment and the data need a clearer explanation. Also, what is to be made of the loss of RPA when MCMBP is absent? And what is the identity of the gray circle next to NME1?

Following your comments, we changed the figure legend and updated the figure. Essentially, we immunoprecipitated MCM3 with or without MCMBP, and then looked at the differences of proteins coprecipitated with MCM3. As shown in the updated Figure 4—figure supplement f, RPA1, FKBP5 and NME1 were top interactors with MCM3 in a

MCMBP dependent manner. We interpreted that RPA1 was lost in the absence of MCMBP because DNA replication became defective (lines 247 to 249). The interaction between MCMBP and FKBP5 has been reported (PMID 25036637) so that we hypothesize FKBP interacted with MCM3 via MCMBP (lines 338 to 342). NME1 (also known as NM23-H1) is a nucleotide diphosphate kinase producing GTP, CTP and TTP, and also works as a ssDNA nicking enzyme (PMID 16818237). Again, this might be related to defective DNA replication without MCMBP.